

# A Comprehensive Global Modelling Assessment of Nitrate Heterogeneous Formation on Desert Dust

Rubén Soussé-Villa[1], Oriol Jorba[1], María Gonçalves Ageitos[1], Dene Bowdalo[1], Marc Guevara[1], and Carlos Pérez García-Pando[1,2]

[1]Barcelona Supercomputing Center, Barcelona, Spain
[2]Catalan Institution for Research and Advanced Studies (ICREA), Barcelona, Spain

**Correspondence:** Rubén Soussé (ruben.sousse@bsc.es), Oriol Jorba (oriol.jorba@bsc)

**Abstract.**

Desert dust undergoes complex heterogeneous chemical reactions during atmospheric transport, forming nitrate coatings that impact hygroscopicity, gas species partitioning, optical properties, and aerosol radiative forcing. Contemporary atmospheric chemistry models show significant disparities in aerosol nitrogen species due to varied parameterizations and inaccuracies in representing heterogeneous chemistry and dust alkalinity. This study investigates key processes in nitrate formation over dust and evaluates their representation in models. We incorporate varying levels of dust heterogeneous chemistry complexity into the MONARCH model, assessing sensitivity to key processes. Our analyses focus on the condensation pathways of gas species onto dust (irreversible and reversible), the influence of nitrate representation on species' burdens and lifetimes, size distribution, and the alkalinity role. Using annual global simulations, we compare particulate and gas species surface concentrations against observations and evaluate global budgets and spatial distributions. Findings show significant outcome dependence on methodology, particularly on the reversible or irreversible condensation of gas species on particles, with a wide range of burdens for particulate nitrate (0.66 to 1.93 Tg) and correlations with observations (0.66 to 0.91). Particulate ammonium burdens display less variability (0.19 to 0.31 Tg). Incorporating dust and sea-salt alkalinity yields results more consistent with observations, and assuming reversible gas condensation over dust, along with alkalinity representation, aligns best with observations, while providing consistent gas and particle partitioning. In contrast, irreversible uptake reactions overestimate coarse particulate nitrate formation. Our analysis provides guidelines for integrating nitrate heterogeneous formation on dust in models, paving the road for improved estimates of aerosol radiative effects.

## 1 Introduction

Desert dust is produced by wind erosion of arid and semi-arid surfaces, contributing approximately 40% of the total dry aerosol mass globally, and between 70% and 80% if sea-salt aerosol (SS) is not considered (Boucher et al., 2013; Adebiyi and Kok, 2020). Dust interacts with solar radiation (direct climate forcing), affects clouds (indirect climate forcing) and alters atmospheric composition, modifying the Earth's energy and water cycles (Pérez et al., 2011; Boucher et al., 2013; Randall et al., 2013). When deposited, dust also affects biogeochemical cycles both on ocean and continental areas (Mahowald et al., 2014; Li et al., 2016; Bergas-Massó et al., 2023). If inhaled, dust can be potentially harmful for animal and human health




(Usher et al., 2003). Temporal variations in dust emissions, from interannual to geological time scales, have been a key driver of the past climate of the Earth, as observed in ice cores and ocean sediment samples (Rea, 1994). All these considerations make desert dust particles a key component of the Earth system influencing climate (Semeniuk and Dastoor, 2020).

Climate perturbations by dust depend fundamentally upon dust particles' physical and chemical properties. These properties are mainly the particle size distribution (PSD), shape, surface characteristics, mineral composition, and mixing state (Usher
et al., 2003; Riemer et al., 2019). These characteristics depend on the dust source region and on its chemical transformations while transported in the atmosphere (Claquin et al., 1999). These two factors determine the final dust optical properties, and consequently its radiative forcing.

Among the factors driving the chemical evolution of dust in the atmosphere, heteregeneous reactions—those involving more than one matter phase—are particularly significant (Schwartz, 1986; Dentener et al., 1996; Bauer et al., 2004, 2007; Riemer
et al., 2019). For example, the condensation of atmospheric gas species on liquid or solid particles are key for particle growth and changes in optical properties during atmospheric transport (Vlasenko et al., 2009; Fairlie et al., 2010; Karydis et al., 2016). Heterogeneous reactions involving nitrogen, for example, can even cause dust to act as a transport medium of nitrates from nitrate-rich areas to regions downwind of dust sources (Ma et al., 2021).

Heterogeneous reactions mainly occur when dust mixes with anthropogenic pollutants emitted in urban and industrial areas.
Nitric acid ($HNO_{3(g)}$), ammonia ($NH_{3(g)}$), sulfur dioxide ($SO_{2(g)}$) and sulfuric acid ($H_2SO_{4(g)}$) are the most important anthropogenic species that react with dust (Usher et al., 2003; Yue et al., 2022). These interactions lead to: 1) the formation of aqueous coatings around the particles (Usher et al., 2003; Krueger et al., 2003, 2004; Fountoukis and Nenes, 2007; Li and Shao, 2009) and 2) the reaction of gases with the non-volatile cations (NVC) present both at the minerals' surfaces and dissolved in the liquid envelope (Dentener et al., 1996; Goodman, 2000; Usher et al., 2003; Li and Shao, 2009). These processes transfer
mass from the gas to the aerosol phase, either through irreversible reactions of low volatile gas vapors with the bulk material of the particle or reversible condensation-evaporation processes between the gas phase and the liquid coating (Usher et al., 2002, 2003; Krueger et al., 2003; Crowley et al., 2010).

Each gas species' chemical transformation follows a distinct pathway depending on its solubility and reactivity with other atmospheric species dissolved in the particle's liquid coating and with the NVC in dust. Among these, $HNO_{3(g)}$ plays a
major role in dust heterogeneous chemistry due to its relatively high solubility and reactivity with ammonium ($NH_4^+$) from dissolved $NH_{3(g)}$, leading to the formation of ammonium nitrate ($NH_4NO_3$) (Metzger et al., 2002; Usher et al., 2003). Aqueous $H_2SO_{4(g)}$ is also paramount, competing with dissolved $HNO_{3(g)}$ to neutralize $NH_4^+$. However, neutralization by $H_2SO_{4(g)}$ predominates due to its very low vapor pressure, preventing its evaporation back to the gas phase and resulting in the formation of ammonium sulfate (($NH_4)_2SO_4$) (Usher et al., 2003; Uno et al., 2020). These are the main formation pathways of particulate
nitrate ($NO_3^-$), particulate ammonium ($NH_4^+$) and particulate sulfate ($SO_4^{2-}$) from the $HNO_3$-$NH_3$-$H_2SO_4$ neutralization system in the particles' liquid coating. Additionally, $SO_{2(g)}$ is relevant in the aqueous medium as it converts to $H_2SO_{4(g)}$ through oxidation with $O_3$ and $H_2O_2$ (Seinfeld and Pandis, 1998; Usher et al., 2002).

The dust minerals also provide additional reactive surfaces to neutralize $HNO_{3(g)}$, with their reactivity depending on their solubility and environmental relative humidity (RH). Under low RH, solid minerals may serve as active sites for reactions



with gas species on the particle surfaces (Usher et al., 2003). Conversely, at high RH, minerals can dissociate in the aqueous
medium, releasing NVC such as $Ca^{2+}$, $K^+$, $Mg^{2+}$ and $Na^+$. The active sites in dust minerals and NVC play a key role
in neutralizing dissolved $HNO_{3(g)}$ and forming $NO_3^-$ compounds ($Ca(NO_3)_{2(a)}$, $Mg(NO_3)_{2(a)}$, $KNO_{3(a)}$ and $NaNO_{3(a)}$)
(Fenter et al., 1995; Krueger et al., 2004). Consequently, these nitrate salts may form on the surface of the dust under low
RH or dissociate in the particle's liquid coating (Usher et al., 2003; Jones et al., 2021). These reactions represent the primary

pathways for $NO_3^-$ formation in the presence of dust, and are highly sensitive to $NO_x$ and sulfate concentrations (Fenter et al.,
1995; Riemer et al., 2003). For instance, the combination of reduced sulfate emissions along with unchanged or even enhanced
$NH_{3(g)}$ emissions, as predicted by 21st century emission scenarios, implies a reduction in particle acidity (Bauer et al., 2007;
Bellouin et al., 2011; Boucher et al., 2013; Hauglustaine et al., 2014; Bian et al., 2017; Karydis et al., 2021). This scenario,
alongside a rise in dust (Usher et al., 2002; Adebiyi et al., 2023), would lead to an increase in $NO_3^-$ formation, especially

in the fine mode, if $NO_x$ emissions are not concurrently reduced (Bauer et al., 2016; Bian et al., 2017; Zaveri et al., 2021).
Therefore, accurate modelling of dust heterogeneous chemistry in atmospheric models is important for present and future air
quality control (Myhre et al., 2006).

In the last decades, several approximations have been introduced in atmospheric chemistry models to address nitrate hetero-
geneous reactions on dust and SS, with a particular focus on $HNO_{3(g)}$ condensation. These approaches range from the dynamic

mass transfer (DMT) calculations between gas and aerosol phases (Meng and Seinfeld, 1996; Lurmann et al., 1997; Song and
Carmichael, 2001; Feng and Penner, 2007; Zaveri et al., 2008; Trump et al., 2015), to the assumption that the bulk gas-aerosol
phases instantly reach thermodynamic equilibrium (TEQ) and calculating their correspondent concentrations (Lurmann et al.,
1997). While DMT can accurately capture processes far from TEQ (e.g. the condensation of gas species at low temperatures,
under extreme relative humidity (RH) conditions or onto coarse particles), the inherent stiffness of inorganic heterogeneous

chemistry renders DMT a rigorous but computationally-expensive methodology (Feng and Penner, 2007; Zaveri et al., 2008;
Trump et al., 2015; Benduhn et al., 2016). On the other hand, assuming instantaneous TEQ is more efficient and has gained
popularity despite its tendency to overestimate coarse nitrate formation (Nenes et al., 1998; Feng and Penner, 2007; Bauer
et al., 2007; Hauglustaine et al., 2014; Paulot et al., 2016; Bian et al., 2017). To balance accuracy and computational cost,
several intermediate strategies have been developed, including 1) simplifying the DMT equations to a first-order irreversible

uptake reaction (UPTK), which ignores the evaporation of uptaken species back to the gas phase (Jacob, 2000; Bauer et al.,
2004; Feng and Penner, 2007; Fairlie et al., 2010), 2) calculating TEQ concentrations for both fine (diameter up to $2.5\mu$m)
and coarse (diameter above $2.5\mu$m) modes of dust and SS (DBCLL), either after kinetically limiting the gas condensing on
each bin or mode (Pringle et al., 2010) or redistributing the condensed mass from the bulk TEQ using kinetic coefficients
(Karydis et al., 2016), and 3) employing a hybrid approach (HYB) that applies TEQ to the fine bins or modes and UPTK to

the coarse ones (Capaldo et al., 2000; Hodzic et al., 2006; Hauglustaine et al., 2014; Trump et al., 2015). Overall, methods
involving DMT or TEQ calculations allow to simulate the reversible heterogeneous reactions (condensation-evaporation dy-
namics), and the UPTK calculates the irreversible uptake of gas species, accounting for gas specifications, particle's alkalinity
and environmental RH (Fairlie et al., 2010; Paulot et al., 2016).



Despite efforts to incorporate nitrate heterogeneous reactions on coarse particles, atmospheric models still significantly

diverge in their predictions of the tropospheric burden of oxidized ($HNO_{3(g)}$+$NO_3^-$) and reduced nitrogen ($NH_{3(g)}$+$NH_4^+$), often struggling to reproduce observational data of these species (Fairlie et al., 2010; Hauglustaine et al., 2014; Paulot et al., 2016; Zakoura and Pandis, 2018; Luo et al., 2019; Jones et al., 2021; Rémy et al., 2022). For instance, the particulate nitrate AeroCom phase III experiment (Bian et al., 2017), an extensive intercomparison study of atmospheric models incorporating $NO_3^-$ formation processes on dust and SS, highlights substantial disagreements. The average $NO_3^-$ atmospheric burden among

models is 0.63 Tg, with a standard deviation of 0.56 Tg, nearly 90% of the mean value. Similar variability is observed for $NH_4^+$ ($0.32 \pm 0.20$ Tg). The study also highlights the general inaccuracy of current models in reproducing observations of $NO_3^-$ concentrations after long-range transport of precursor species, indicating that nitrogen heterogeneous chemistry processes on dust and SS are often misrepresented.

In this work, we systematically investigate the underlying processes governing heterogeneous chemistry that lead to $NO_3^-$

formation. We study the pathways driving nitrate formation on dust. To achieve this goal, we incorporate a variety of mechanisms of different complexities into a global model. This enables a comprehensive analysis of the partitioning between gas and aerosol phases, the suitability of irreversible and reversible parameterizations for the gas species condensation on coarse dust, and the role of explicit representation of alkalinity. While our primary emphasis is on the heterogeneous chemistry on dust surfaces, we also account for nitrate formation on SS and its alkalinity.

This paper is structured as follows. Section 2 introduces the MONARCH atmospheric chemistry model, detailing the specific developments implemented for this study (2.2), the setup of the simulations conducted (2.4) and the datasets used for evaluation (2.5). Section 3 presents an analysis of the global simulations and their evaluation against observational data. This section includes a comparison of the spatial distributions and an examination of the total nitrogen burden and its gas/particle partitioning. Additionally, we discuss the budgets of reduced and oxidized nitrogen species, depositions, production/loss rates

and lifetimes. Our results are contextualized with findings from previous studies, providing a comprehensive understanding of results. Section 4 provides a summary of our key findings.

## 2 Methods

### 2.1 The MONARCH model

MONARCH is an atmospheric chemistry model developed at the Earth Sciences department of the Barcelona Supercomputing

Center (Pérez et al., 2011; Jorba et al., 2012; Badia et al., 2017; Klose et al., 2021; Gonçalves Ageitos et al., 2023; Navarro-Barboza et al., 2024). It simulates the life cycle of aerosol and gas-phase species in the atmosphere, utilizing an online coupling with the Nonhydrostatic Multiscale Model on the B-grid (NMMB) (Janjic and Gall, 2012). NMMB allows running both global and regional domains with embedded telescoping nests. The Arakawa B grid is used in the horizontal direction and the Lorenz hybrid pressure-sigma coordinate in the vertical direction. MONARCH works on a latitude-longitude discretization with polar

filtering and a rotated longitude-latitude grid is adopted for regional applications. The NMMB numerical schemes are based on principles described in Janjic and Gall (2012). The physical parameterizations used in the model include (1) a surface



layer scheme based on the Monin-Obuknov similarity theory (Monin and Obukhov, 1954) combined with a viscous sublayer on continental and water surfaces (Zilitinkevich, 1965; Janjic, 1984, 1996), (2) the Mellor–Yamada–Janjic (MYJ) planetary boundary layer and the free troposphere turbulence scheme (Janjić, 2001), (3) the Unified NCEP-NCAR-AFWA Noah land

surface model (Ek et al., 2003) to compute the surface heat and moisture fluxes, (4) the 1D Rapid Radiative Transfer Model for Global circulation Models (RRTMG) (Iacono et al., 2008) for the calculation shortwave and longwave radiative fluxes, (5) the Ferrier microphysics scheme (Ferrier et al., 2002) for grid-scale clouds, and (6) the Betts–Miller–Janjic convective clouds scheme (Betts and Miller, 1986; Emanuel and Živković Rothman, 1999; Janjić, 2000). The same advection and vertical mixing schemes formulated in NMMB are used for both meteorological and chemistry species for consistency.

MONARCH includes a gas-phase module combined with a hybrid sectional–bulk multi-component mass-based aerosol module. The gas-phase chemistry is based on the Carbon Bond 2005 (CB05) chemical mechanism extended with chlorine chemistry (Yarwood et al., 2005; Whitten et al., 2010) designed to describe urban to remote tropospheric conditions. The photolysis rates are computed using the Fast-J scheme (Wild et al., 2000) accounting for aerosols, clouds, and absorbers such as ozone. A resistance approach is adopted for dry deposition (Wesely, 1989) and in-cloud and below-cloud scavenging and

wet deposition follows Byun (1999) and Foley et al. (2010).

    The aerosol representation in MONARCH considers 8 main components, namely dust, SS, black carbon, organic matter (both primary and secondary), $SO_4^{2-}$, $NH_4^+$, $NO_3^-$ and unspeciated aerosol mass. Mineral dust and SS are described with a sectional size distribution of 8 bins spanning from $0.2 \mu m$ to $20 \mu m$ and $30 \mu m$ in diameter, respectively. All the other aerosol components are represented by a fine mode, except $NO_3^-$ which is extended with a coarse mode to consider the condensation

of $HNO_{3(g)}$ on coarse particles. Table S2 in the Supplementary material reports the bin volumetric and effective radii, density and the bin or mode fractional contribution to $PM_{2.5}$ and $PM_{10}$ for $SO_4^{2-}$, $NH_4^+$, $NO_3^-$. Black carbon is considered in two primary hydrophobic/hydrophilic modes and emitted as 80% hydrophobic. An ageing process during its transport with an e-folding of 1.2 days transfers mass from hydrophobic to hydrophilic modes (Chin et al., 2002). Organic aerosols follow the simple scheme of Pai et al. (2019) assuming fixed secondary organic aerosol (SOA) yields adjusted to match more complex

approaches (i.e. volatility based schemes). Similar to black carbon, primary organic emissions are emitted as 50% hydrophobic with an ageing e-folding of 1.15 days. Designed for global models, this approach has also shown good results at regional scale (Navarro-Barboza et al., 2024).

    A simplified gas–aqueous–aerosol mechanism accounts for sulfur chemistry through the oxidation of $SO_{2(g)}$ and dimethyl sulfide (DMS) and assuming complete nucleation for the remaining $H_2SO_{4(g)}$ as $SO_4^{2-}$ after the chemistry integration time

step (Spada, 2015). The heterogeneous hydrolysis of $N_2O_5$ on aqueous sulfate particles accounts for additional $HNO_{3(g)}$ formation following the formulation of Riemer et al. (2003). Prior to this study, secondary nitrate–ammonium aerosol was solved using TEQ model EQuilibrium Simplified Aerosol Model version v03b (EQSAM v03b, Metzger et al. (2002)) for fine particles. Note that EQSAM v03b only considers the partitioning of sulfate-nitrate-ammonium without considering the presence of other species (i.e. Dust or SS alkalinity). (Below we describe the adoption of ISORROPIA-II v1 (Milousis et al.,

2024) for the purpose of this study.) To account for secondary nitrate aerosol formation on coarse dust and SS particles, an





hybrid (HYB) approach is employed through an uptake reaction (UPTK) of $HNO_{3(g)}$. The reaction uses the uptake rate ($K$) defined by Jacob (2000) as a first-order function (Schwartz, 1986):

$$K = \left( \frac{r}{D_g} + \frac{4}{v\gamma} \right)^{-1} \cdot S \qquad (1)$$

where $r$ is the aerosol bin radius, $D_g$ is the gas-phase diffusion coefficient, $v$ the mean molecular speed, $S$ the aerosol specific surface area, and $\gamma$ the uptake coefficient, defined as the ratio of the number of gas molecules reacting with the particle's surface over the fraction of molecules being absorbed by the given surface (i.e. the accommodation coefficient) (Phadnis and Carmichael, 2000; Guimbaud et al., 2002). A $\gamma$ value of 0.1 is assumed for dust (Hanisch and Crowley, 2001; Vlasenko et al., 2006) and 0.01 for SS (Tolocka et al., 2004). The production of fine and coarse $NO_3^-$ is traced in separated bins.

Finally, MONARCH includes meteorology-driven emission modules for key species. Emissions of biogenic non-methane volatile organic compounds (NMVOC) and NO are calculated from the Model of Emissions of Gases and Aerosols from Nature (MEGAN) v2.04 (Guenther et al., 2006). Several SS source functions are available in the model (Spada et al., 2013); here we use the Jaeglé et al. (2011) formulation. Similarly, different parameterizations for dust emissions are available ranging from more simplified to more physics-based descriptions (Klose et al., 2021). Following Gonçalves Ageitos et al. (2023), the G01-UST scheme described in Klose et al. (2021) is used in this work. For dust emission, the topography-based source function from Ginoux et al. (2001) is used to scale the dust flux from available sources and to set its dependency on surface friction velocity. Dust emission is limited to areas presenting a frequency of occurrence of dust optical depth above 0.2, identified using maps created from Moderate Resolution Imaging Spectroradiometer (MODIS) Deep Blue retrievals (Hsu et al., 2004; Ginoux et al., 2012). Surface roughness influence on dust emission is parameterized based on Raupach et al. (1993), whose vegetation cover is determined using surface reflectance from Landsat and MODIS monthly data (Raupach et al., 1993; Guerschman et al., 2015).

## 2.2 Model updates

In this study, we investigate the primary chemical pathways responsible for $NO_3^-$ formation on coarse particles by integrating mechanisms of varying complexity within the global model MONARCH (Figure 1). Table 1 lists the irreversible and reversible heterogeneous reactions considered in our analysis. Here, we detail the enhancements implemented in MONARCH to address partially or fully the array of reactions of interest, with a primary focus on maintaining a balance between complexity, accuracy, and computational efficiency in the resulting solution.

To trace the formation of fine and coarse $NO_3^-$, $NH_4^+$ and $SO_4^{2-}$, an additional hydrophilic bin for the coarse mode of these three species is added to the default MONARCH size parameterization, as reported in the Supplementary Table S2.

### 2.2.1 Irreversible heterogeneous chemistry of nitrate and sulfate

A widely adopted method to simulate $NO_3^-$ and $SO_4^{2-}$ formation on coarse particles involves incorporating irreversible heterogeneous reactions of gas species on dust and SS through a first-order uptake parameterization. Specifically, the uptake of





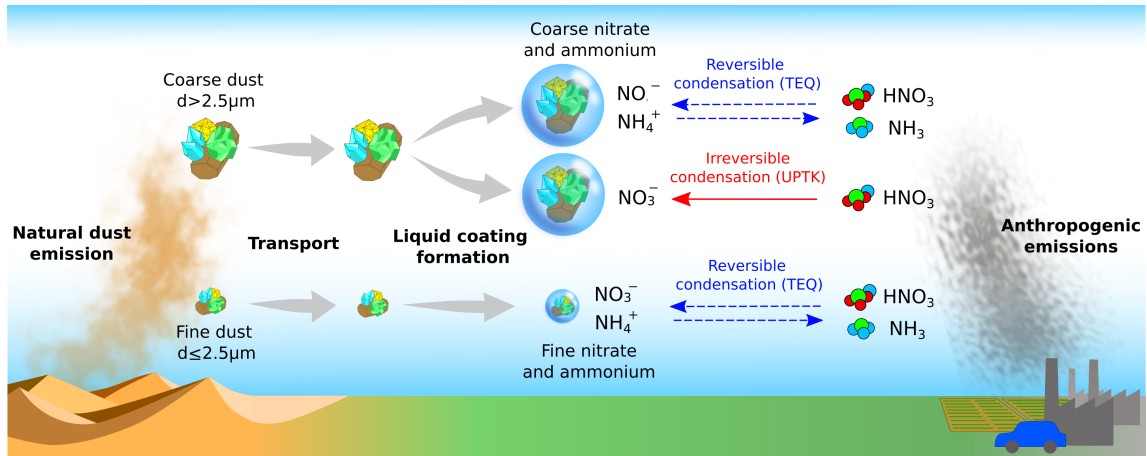

**Figure 1.** Graphical illustration of the respective mineral dust heterogeneous chemistry mechanisms implemented in this work for fine and coarse particles. Fine dust aqueous coating is assumed to reach TEQ with anthropogenic gas species to form fine particulate nitrate ($NO_3^-$). Conversely, coarse particulate nitrate is formed either through the reversible condensation (TEQ) of $HNO_{3(g)}$ and $NH_{3(g)}$ after kinetic limitation, or through the irreversible uptake reaction (UPTK) of $HNO_{3(g)}$ on coarse dust particles. Particulate ammonium ($NH_4^+$) is formed in both modes if TEQ is assumed.

$HNO_{3(g)}$ on coarse particles is commonly assumed to drive coarse $NO_3^-$ formation (Jacob, 2000; Hodzic et al., 2006; Bauer et al., 2004; Feng and Penner, 2007; Fairlie et al., 2010; Hauglustaine et al., 2014; Paulot et al., 2016; Jones et al., 2021), while the uptake of $SO_{2(g)}$ is known to lead to the formation of coarse $SO_4^{2-}$ (Phadnis and Carmichael, 2000; Song and Carmichael, 2001; Usher et al., 2002; Prince et al., 2007; Fairlie et al., 2010; Li et al., 2012; Liu and Abbatt, 2021; Yue et al., 2022). Most

models assume constant uptake coefficients ($\gamma$) for these reactions, for example $\gamma = 0.1$ for $HNO_{3(g)}$ uptake on dust (Dentener et al., 1996; Hanisch and Crowley, 2001, 2003; Bauer et al., 2004; Hodzic et al., 2006; Wei, 2010)). However, recent studies have shown that using this value tends to overestimate $NO_3^-$ formation (Vlasenko et al., 2006; Mashburn et al., 2006; Fairlie et al., 2010). This suggests that the uptake coefficient for $HNO_{3(g)}$ should be lower and that it is highly influenced by RH

(Goodman, 2000; Krueger et al., 2003; Vlasenko et al., 2006, 2009; Fairlie et al., 2010; Wei, 2010) and dust alkalinity (Goodman, 2000; Hanisch and Crowley, 2001; Krueger et al., 2004; Liu et al., 2007; Prince et al., 2007; Wei, 2010; Crowley et al., 2010). Recent studies increasingly implement $\gamma$ as a function of RH and employ different parameterizations of this function to account for dust alkalinity.

For our study, we extended the chemical mechanism of MONARCH to incorporate pathways for the formation of coarse

$NO_3^-$ and sulfate aerosols (Figure 2). This extension involved refining irreversible heterogeneous parameterizations within the model, specifically the uptake of $HNO_{3(g)}$ on dust and SS particles (R4-5 in Table 1), as well as the uptake of $SO_{2(g)}$ on dust particles (R3 in Table 1). Our implementation incorporates dependencies of $\gamma$ on RH and the alkalinity of dust (Vlasenko et al., 2006, 2009; Liu et al., 2007, 2008; Crowley et al., 2010; Fairlie et al., 2010; Wei, 2010).





**Table 1.** Heterogeneous reactions implemented in MONARCH. Dust and sea-salt particles are referred as *DU* and *SS*, respectively.

| Irreversible reactions | | Reaction | Process | Notes |
|---|---|---|---|---|
| | (R1) | $DU_{(aer)} + H_2SO_{4(g)} \rightarrow SO_{4(aq)}^{2-}$ | condensation | 1 |
| | (R2) | $SS_{(aer)} + H_2SO_{4(g)} \rightarrow SO_{4(aq)}^{2-}$ | condensation | 1 |
| | (R3) | $DU_{(aer)} + SO_{2(g)} \rightarrow SO_{4(aer)}$ | UPTK ($\gamma_1$) | 2 |
| | (R4) | $CaCO_{3(aer)} + 2HNO_{3(g)} \rightarrow Ca(NO_3)_{2(aer)} + H_2O + CO_2$ | UPTK ($\gamma_2$) | 3,4 |
| | (R5) | $SS_{(aer)} + HNO_{3(g)} \rightarrow NaNO_{3(aer)} + HCl_{(g)}$ | UPTK ($\gamma_3$) | 4,5 |

| Gas-particle equilibrium reactions | | Reaction | Process | Notes |
|---|---|---|---|---|
| | (R6) | $HNO_{3(g)} + NH_{3(g)} \rightleftharpoons NH_{4(aq)}^+ + NO_{3(aq)}^-$ | TEQ | 6 |
| | (R7) | $H_2SO_{4(g)} + NH_{3(g)} \rightleftharpoons NH_{4(aq)}^+ + SO_{4(aq)}^{2-}$ | TEQ | 6,7 |
| | (R8) | $CaCO_{3(aq)} + 2HNO_{3(g)} \rightleftharpoons Ca_{(aq)}^{2+} + 2NO_{3(aq)}^- + H_2O + CO_2$ | TEQ | 4,8 |
| | (R9) | $MgCO_{3(aq)} + 2HNO_{3(g)} \rightleftharpoons Mg_{(aq)}^{2+} + 2NO_{3(aq)}^- + H_2O + CO_2$ | TEQ | 4,8 |
| | (R10) | $K_2CO_{3(aq)} + 2HNO_{3(g)} \rightleftharpoons 2K_{(aq)}^+ + 2NO_{3(aq)}^- + H_2O + CO_2$ | TEQ | 4,8 |
| | (R11) | $Na_2CO_{3(aq)} + 2HNO_{3(g)} \rightleftharpoons 2Na_{(aq)}^+ + 2NO_{3(aq)}^- + H_2O + CO_2$ | TEQ | 4,8 |
| | (R12) | $NaCl_{(aq)} + HNO_{3(g)} \rightleftharpoons Na_{(aq)}^+ + NO_{3(aq)}^- + HCl_{(g)}$ | TEQ | 4,9 |

1. Sulfuric acid ($H_2SO_{4(g)}$) is assumed to completely condense on fine and coarse dust and SS, transferring mass to the respective size mode. The condensation reaction does not depend on dust and SS alkalinity, but solely on their specific surface area based on Pringle et al. (2010).

2. Sulfur dioxide ($SO_{2(g)}$) uptake coefficient on dust ($\gamma_1$) is a function of RH as defined in Fairlie et al. (2010) and based on experimental studies performed on calcite particles by Prince et al. (2007). However, to account for $SO_{2(g)}$ oxidation by deliquesced $O_3$ and $NO_2$, the UPTK reaction of $SO_{2(g)}$ is performed even in the absence of alkalinity. It assumes same dust alkalinity as in nitric acid uptake (R4).

3. Nitric acid ($HNO_{3(g)}$) uptake coefficient on $CaCO_{3(aq)}$ ($\gamma_2$) is a function of RH as defined by Fairlie et al. (2010) based on experimental studies performed on calcite particles by Liu et al. (2008). $CaCO_{3(aq)}$ concentration is used instead of $DU_{(aer)}$ concentration because the uptake coefficient is scaled for alkalinity as shown in equation 2.

4. $CaCO_{3(aq)}$, $MgCO_{3(aq)}$, $K_2CO_{3(aq)}$, $Na_2CO_{3(aq)}$ and $NaCl_{(aq)}$ refers to the NVC content derived from the $DU_{(aer)}$ and $SS_{(aer)}$ concentrations using fractions from Gonçalves Ageitos et al. (2023) for dust and from Seinfeld and Pandis (2006) for SS (Section 2.2.3).

5. $HNO_{3(g)}$ uptake coefficient on SS ($\gamma_3$) is based on the experimental study Liu et al. (2007), that report $HNO_{3(g)}$ uptake kinetics for different RH and sea-salt particles size. The average values $\gamma_3 = 0.15$ for particles between 1 and 2.5 $\mu m$ and $\gamma_3 = 0.05$ for particles from 2.5 to 10 $\mu m$ are assumed, considering ambient RH=80%.

6. Neutralization of $HNO_{3(g)}$ and $H_2SO_{4(g)}$ by ammonia is calculated through TEQ with ISORROPIA-II in the fine mode and additionally in the coarse mode if coarse $NO_3^-$ and $NH_4^+$ formation is considered (Myhre et al., 2006; Usher et al., 2003; Uno et al., 2020). All reactants are assumed to remain in the aqueous phase (metastable assumption).

7. The result of the neutralization of $H_2SO_{4(g)}$ can be $(NH_4)_2SO_{4(aer)}$, $NH_4HSO_{4(aer)}$ or $(NH_4)_3H(SO_4)_{2(aer)}$ if solid results were assumed (Liu et al., 2022), but under metastable assumption only aqueous ions of $SO_4^{2-}$ are considered.

8. Calcium, magnesium, potassium and sodium (NVC) deliquesced from carbonates present in the dust particles' bulk neutralize $HNO_{3(g)}$ in the liquid coating of dust particles (Usher et al., 2003; Krueger et al., 2004; Fountoukis and Nenes, 2007; Hauglustaine et al., 2014). Dust NVC content (i.e. alkalinity) is dependent on particle size and globally averaged from Journet et al. (2014) mineral data.

9. Sodium chloride from sea-salt particles dissolves and reacts with $HNO_{3(g)}$ in the liquid coating of sea-salt particles (Myhre et al., 2006). Sea-salt also presents other NVC that are included in reactions R8-11, which are assumed globally homogeneous from Seinfeld and Pandis (2006) and Karydis et al. (2016).





The $\gamma$ dependency on RH is modeled akin to a Brunauer–Emmett–Teller (BET) isotherm, which characterizes water adsorption on dust particles (Vlasenko et al., 2006). We employed a modified BET function to formulate $\gamma$, extending it to account for dust alkalinity. This formulation is represented by the following equation:

$$\gamma = Sc \cdot \frac{|c_1 \cdot RH|}{(1 - RH)|1 + c_2 \cdot RH|} \tag{2}$$

where RH is the relative humidity (ranging from 0 to 1), $c_1$ and $c_2$ denote the water adsorption scaling factors (Vlasenko et al., 2006), and $Sc$ is a factor dependent on dust alkalinity. For the uptake of $HNO_{3(g)}$ on dust (R4-5 in Table 1), typical values assumed for $c_1$ and $c_2$ are 8 and 7, respectively (Li et al., 2012; Paulot et al., 2016; Wang et al., 2017). However, literature reports varying values for $Sc$ based on dust alkalinity assumptions, ranging from $Sc = \frac{1}{30}$ for the industrially-standardized Arizona Test Dust (Möhler et al., 2006; Herich et al., 2009; Suman et al., 2024) to $Sc = 0.018$ for samples from the China Loess (with 39% $CaCO_3$ content) (Krueger et al., 2004; Wei, 2010).

We adopt the uptake RH functions for both $HNO_{3(g)}$ and $SO_{2(g)}$ on dust from Fairlie et al. (2010). To fit experimental data from Song et al. (2007) and the RH function reported by Fairlie et al. (2010), we determine $c_1 = 3.84 \cdot 10^{-4}$ and $c_2 = 0.56$ for $\gamma(HNO_3)$. Additionally, Fairlie et al. (2010) assumes a NVC content of 3.0% of Ca and 0.6% of Mg, which differs from the NVC values used in the present study. Therefore, we use the alkalinity scaling factor $Sc$ to normalize the Fairlie et al. (2010) function accordingly, as described in Section 2.2.3. Specifically, values for $Sc$ are defined as the ratio of Ca and Mg percentage used in our study relative to those assumed by Fairlie et al. (2010), resulting in $Sc = 1.80$ and $Sc = 1.52$ for the two alkalinity values used in our experiments (see Section 2.2.3 below). Note that $Sc$ is zero if no alkalinity is considered and that a constant value of $\gamma(HNO_3) = Sc \cdot 1.05 \cdot 10^{-3}$ is used for RH higher than 80%.

Similarly, we determine incorporating the $\gamma(SO_2)$ on dust (R3 in Table 1) fitting equation 2 to experimental data from Prince et al. (2007) and the RH function from Fairlie et al. (2010), yielding values of $c_1 = 2.7 \cdot 10^{-6}$ and $c_2 = -1.06$. The same $Sc$ as used for $\gamma(HNO_3)$ is applied here. However, if alkalinity is not considered $Sc$ is set to 1.0 (and not zero) to account for the oxidation of $SO_{2(g)}$ by deliquesced $O_3$ and $NO_2$ (Usher et al., 2002; Prince et al., 2007). For RH above 90%, $\gamma(SO_2)$ remains constant at $\gamma(SO_2) = Sc \cdot 5.0 \cdot 10^{-4}$.

For the $HNO_{3(g)}$ uptake on SS (R5 in Table 1), we adopted the $\gamma(HNO_3)$ values of Liu et al. (2007), which provide experimental estimates of this factor for different particle sizes and RH. However, a clear uptake function on these parameters has been found in the literature. Therefore, for the sake of simplicity, we defer an implementation of an uptake coefficient dependent on these metrics to future research. In this study, we did not account for the RH dependency of $\gamma(HNO_3)$, and instead we used average values at 80% RH resulting in $\gamma(HNO_3) = 0.15$ for SS particles in the range of 0.1 to 2.5 $\mu$m and $\gamma(HNO_3) = 0.05$ for particles larger than 2.5 $\mu$m. While larger values have been reported by Guimbaud et al. (2002), we opted for these values as they align with more widely accepted ranges found in the literature (Saul et al., 2006; Pratte and Rossi, 2006; Liu et al., 2007; Fagerli et al., 2015). Using higher values could potentially overestimate the uptake on SS particles.

The condensation of $H_2SO_{4(g)}$ (R1-2 in Table 1) on dust and SS is another relevant source of $SO_4^{2-}$ introduced in our model. Due to the extremely low volatility of $H_2SO_{4(g)}$ at atmospheric temperatures, its condensation onto existing particles



is assumed as irreversible and nearly complete. This process signifies a direct mass transfer from the gas to the aerosol phase (Zaveri et al., 2008; Hauglustaine et al., 2014). The amount of $H_2SO_{4(g)}$ that condenses in the fine and coarse modes is determined using kinetic diffusive coefficients calculated as described in section 2.3.

### 245  2.2.2   Reversible heterogeneous chemistry of nitrate and ammonium

The gas-aerosol partitioning of semivolatile inorganic aerosols in previous studies with MONARCH was based on the EQSAM v03b TEQ model. EQSAM provides a computationally efficient approach that bypasses the expensive iterative activity coefficient calculation employed in other thermodynamic models. EQSAM was originally designed to handle the partitioning of ammonium-sulfate-nitrate-water system excluding solid components and was extended to include solids, HCl and Cl-/Na+ in
the version v03b, the one used in MONARCH. One of the limitations of such version is the lack of information on NVC and/or mineral species in traced species.

For this study, we implemented the ISORROPIA-II v1 (Fountoukis and Nenes, 2007) TEQ model as an additional option in MONARCH to investigate the sensitivity of the partitioning of semivolatile inorganic compounds to NVCs. While a more recent version of ISORROPIA-II (v2.3) exists, which improves aerosol pH estimations at near pH-neutral conditions (Song
et al., 2018), global scale simulations have shown only minor differences when compared to ISORROPIA-II v1 (Milousis et al., 2024).

ISORROPIA-II v1 determines TEQ concentrations of gas, liquid and solid phases. It can assume either stable conditions, where compounds precipitate into solids, or metastable conditions, where compounds remain as supersaturated liquid solutions. To enhance computational efficiency, ISORROPIA-II employs a segmented approach for calculating TEQ concentrations. This
approach defines five different regimes based on the ratios of precursor species (i.e. sulfate, sodium and crustal species), RH and temperature. Each regime addresses a specific subset of relevant species and equilibrium equations. Efficiency is further improved by retrieving species' activity coefficients from look-up tables (Fountoukis and Nenes, 2007; Milousis et al., 2024). The medium acidity is determined by the concentrations of acidic/basic gaseous species ($HNO_{3(g)}$, $NH_{3(g)}$, $H_2SO_{4(g)}$), particle ($NH_4^+$, $SO_4^{2-}$, $NO_3^-$), and crustal ions ($K^+$, $Ca^{2+}$, $Mg^{2+}$, $Na^{2+}$, $Cl^-$), which are inputs to ISORROPIA-II. After TEQ
is calculated with these species, the resulting pH is provided by the thermodynamic model.

In this work, we use the metastable solution of ISORROPIA-II, assuming all the resulting particulate compounds from TEQ computation to remain in the liquid phase. Previous studies comparing stable and metastable methodologies with ISORROPIA-II have reported only marginal differences in global nitrate budgets between both modes. These differences noted slightly lower pH values and nitrate formation when using the metastable assumption (Karydis et al., 2016, 2021; Milousis et al., 2024).
Furthermore, the metastable assumption allows for full traceability of total aerosol nitrate, ammonium and sulfate formation (reactions R5-8 in Table 1).

We also adopt the temperature and pressure applicability range for ISORROPIA-II proposed by Sulprizio (2022), which highlights potential instabilities in reactions occurring below 250ºK and 200hPa. Consequently, ISORROPIA-II computations are limited to cells with temperature and pressure values above these thresholds.





### 2.2.3 Dust and sea-salt alkalinity

Alkalinity refers to the ability of a substance to neutralize acids and maintain a stable pH level. Both dust and SS particles contain NVCs that contribute to the overall alkalinity of the aerosol, thereby neutralizing gas acidic species such as $HNO_{3(g)}$ and sulfates.

To investigate the importance of representing dust and SS alkalinity, we derive a global average size-dependent NVCs content from 5-year long MONARCH simulations that explicitly track dust mineral species (Gonçalves Ageitos et al., 2023). To assess the uncertainty arising from our limited knowledge of the soil mineralogy of dust sources, we relied on two different MONARCH experiments detailed in (Gonçalves Ageitos et al., 2023), which utilized the Claquin et al. (1999) and Journet et al. (2014) soil mineralogical datasets, respectively. The simulation based on Claquin et al. (1999) accounts for eight distinct minerals, whereas the simulations based on Journet et al. (2014) for twelve minerals (Table S3). The Claquin et al. (1999) dataset includes quartz, feldspar, illite, smectite, kaolinite, calcite, gypsum, and hematite. The Journet et al. (2014) dataset includes these minerals as well as chlorite, vermiculite, mica, and goethite (Table S5). In this study, we adopt an upper bound for the minerals' solubility and reactivity with gas species based on Hanisch and Crowley (2001). Moreover, we assume size-dependent but globally homogeneous values for dust mineralogy, and therefore dust alkalinity and NVC, to focus on understanding heterogeneous reaction parameterizations. We defer the analysis of the potential importance of dust mineralogical variations upon dust heterogeneous chemistry to a forthcoming study.

Based on the global average mineral mass fraction for each dust size bin derived from the mineralogy simulations and the elemental composition associated to each mineral (see the Supplementary Tables S4 and S6) we estimate the average NVC content per dust size bin at each time step following equation 3:

$$NVC_{i,j} = \left( \sum_k fNVC_{k,j} \cdot M_{i,k} \right) \cdot DU_i \tag{3}$$

where the NVC concentration for each element $j$ (i.e., Ca, Mg, K, Na) and size bin $i$ ($NVC_{i,j}$) at a given location is derived by considering the NVC molar fraction in each mineral $k$ ($fNVC_{k,j}$), the global average mass fraction of each mineral and size bin ($M_{i,k}$) and the bin's dust concentration at the specified location ($DU_i$). $NVC_{i,j}$ serves as input to the TEQ calculation. We consider only minerals soluble in water or acids that may at least partly dissolve in the liquid coating of the particles (Usher et al., 2003), and only NVC reacting with the gas species in ISORROPIA-II are used (calcite, magnesium, potassium, sulfate, chlorite or sodium) for the calculation.

The dust NVC global average content result in: 5.17% $Ca^{2+}$, 0.79% $Na^+$, 2.37% $K^+$, 1.32% $Mg^{2+}$ for the Journet et al. (2014) dataset, and 3.68% $Ca^{2+}$, 0.87% $Na^+$, 3.15% $K^+$, 1.75% $Mg^{2+}$ for Claquin et al. (1999). The size-resolved NVC percentages for each dust bin are reported in the Supplementary Tables S3, S4, S5 and S6. These values within the range of Karydis et al. (2016) (5.36±3.69% $Ca^{2+}$, 2.46±1.90% $Na^+$, 2.08±1.34% $K^+$, 1.96±2.20% $Mg^{2+}$).





Additionally, as discussed in section 2.2.1, the derived dust NVC implies the application of the scaling factors $Sc = 1.80$ and $Sc = 1.52$ in eq 2 for the irreversible uptake experiments, assuming the average alkalinity derived from the Journet et al. (2014) and Claquin et al. (1999) simulations, respectively.

Regarding SS, we use a global average composition from Seinfeld and Pandis (2006) with 55% $Cl^-$, 30.6% $Na^+$, 7.7% $SO_4^{2-}$, 3.7% $Mg^{2+}$, 1.2% $Ca^{2+}$, and 1.1% $K^+$.

**2.3   Nitrate mechanisms under study**

In atmospheric conditions, $HNO_{3(g)}$ and $NH_{3(g)}$ do not exhibit as low volatility as $H_2SO_{4(g)}$. Consequently, their condensation onto liquid coatings around particles is a reversible process and should not be assumed as irreversible uptake reactions (Usher et al., 2003). Different mechanisms have been proposed to model the partitioning of nitrate and ammonium across the entire aerosol size range, aiming to mitigate the computationally expensive cost of solving the dynamic mass transfer equations
(Capaldo et al., 2000; Feng and Penner, 2007; Hauglustaine et al., 2014).

The assumption of TEQ between gas and aerosol phases provides a practical approximation to account for the potential evaporation of already dissolved molecules in the liquid coating of fine particles. Equilibrium timescales for fine ammonium nitrate (diameter less than $1\mu m$) are typically on the order of minutes under typical atmospheric conditions (Wexler and Seinfeld, 1990; Dassios and Pandis, 1999). However, in TEQ models, it is assumed that TEQ is reached within each model
time-step (on the order of few minutes). This assumption is reasonable for fine particles, but less so for coarse particles, where achieving equilibrium can take minutes to hours (Feng and Penner, 2007).

To overcome this limitation, different approaches have been proposed in the literature to incorporate the condensation-evaporation of $HNO_{3(g)}$ and $NH_{3(g)}$ on coarse particles while minimizing computational costs. In this study, we explore two such methods. The hybrid method (HYB) (Hodzic et al., 2006; Feng and Penner, 2007) solves the partitioning over fine particles
using a TEQ model and employs a first-order irreversible uptake (UPTK) reaction for condensation over coarse particles (section 2.2.1). More refined approaches treat the formation of coarse $NO_3^-$ as a reversible process through the combination of a double call to TEQ calculation, one for the fine and one for the coarse mode, together with a kinetic limitation of the gas species involved in the partitioning (DBCLL) (Pringle et al., 2010). These two mechanisms are evaluated in this study to assess their impact on the formation of coarse $NO_3^-$ and $NH_4^+$, as illustrated in Figure 2. Additionally, for the purpose of comparison,
a scheme neglecting coarse $NO_3^-$ formation (fTEQ) is also employed. We briefly describe each approach below.

To assess the effect of neglecting coarse $NO_3^-$ and $NH_4^+$ formation on atmospheric composition, the fTEQ approach solves the partitioning of semivolatile inorganic species with ISORROPIA-II exclusively within the fine mode. This mechanism solves the nitric-ammonia-sulfate neutralization (R6-7 in Table 1), accounting for alkalinity effects (R8-12 in Table 1) if fine dust and SS are considered in the mixture. Since $H_2SO_{4(g)}$ and $SO_4^{2-}$ influence the ambient pH, they are always involved in
any TEQ calculation. As described in Section 2.1, aqueous sulfate formation is solved through the oxidation of $SO_{2(g)}$ and DMS. Here, however, only 50% of the remaining $H_2SO_{4(g)}$ is assumed to directly nucleate as fine $SO_4^{2-}$ through the aqueous phase chemistry, while the rest condenses into fine $SO_4^{2-}$ through reactions R1 and R2 in Table 1 (Figure 2 (a)). While in our study fTEQ serves as a sensitivity test to assess the impact of neglecting coarse $NO_3^-$ and $NH_4^+$ formation, the fTEQ approach





| | t0: Fine SO4 | t1: Fine DU | t2: Coarse DU | t3: Gas HNO3, NH3 and H2SO4 | t4: Fine DU & SS | t5: Coarse DU | t6: Coarse SS |
|---|---|---|---|---|---|---|---|
| **fTEQ** | Cond. (x1.0 SO4) | UPTK (SO2) | UPTK (SO2) | - | TEQ (NH3 & HNO3) | - | - |
| **HYB** | | | | - | | UPTK (HNO3) | UPTK (HNO3) |
| **DBCLL** | Cond. (x0.5 SO4) | UPTK (SO2) | | DIFFLIM | | TEQ (NH3 & HNO3) | |

**Table 2.** Sequence of reactions and calculations performed for each scheme (rows). In columns, the order (t$N$) and the mode (gas/aerosol) for each process is indicated. Dust and sea-salt particles are referred as *DU* and *SS*, respectively.

may be appropriate in environments where coarse particles are sparse or in applications focusing primarily on fine particle
formation (Bian et al., 2017).

    Conversely, the HYB mechanism (Figure 2 (a)) solves $NO_3^-$ formation through a sequential implementation of 1) TEQ reaction between $HNO_{3(g)}$ and $NH_{3(g)}$ considering internal mixing with fine dust and SS modes (R6-12 in Table 1), and 2) an irreversible first-order UPTK reaction of the remaining $HNO_{3(g)}$ on the coarse modes of dust and SS (R4 and R5 in Table 1). The UPTK reaction of $HNO_{3(g)}$ follows the implementation detailed in section 2.2.1. Sulfate is treated in an analogous
manner to the fTEQ mechanism.

    Finally, the DBCLL mechanism (Figure 2 (b)) (Pringle et al., 2010) treats coarse $NO_3^-$ and $NH_4^+$ formation as a reversible condensation-evaporation process. Firstly, the DBCLL methodology involves the calculation of kinetic diffusion limitation coefficients (DIFFLIM) for both the fine and coarse size modes of each of the condensing gas species, which restricts the amount of gas available to condense on each mode (Table 2). DIFFLIM has been implemented based on the formulation by
Vignati et al. (2004) for $H_2SO_{4(g)}$, and its extension to other gases by Pringle et al. (2010). Following DIFFLIM, sequential TEQ calculations are conducted over fine and coarse modes (double call of the TEQ model, reactions R8-12 in Table 1), using the DIFFLIM coefficients to limit the availability of gas condensing on each mode.

    Regarding sulfate treatment in DBCLL, as similar approach to fTEQ is employed, but the DIFFLIM coefficients calculated for $H_2SO_{4(g)}$ are used to condense the available $H_2SO_{4(g)}$ into either fine or coarse modes of $SO_4^{2-}$ (reactions R1 and R2 in
Table 1 and Figure 2 (b)).

## 2.4   Experimental setup

We performed global simulations based on the mechanisms described in Section 2.3 (Table 3). Overall, twelve different runs are analyzed to test different degrees of complexity and sensitivity to parameterizations when simulating heterogeneous chemistry of dust and SS, such as the hypothesis on reversibility on nitrate formation, and the role that dust and SS alkalinity play in the
partitioning of gas and aerosol species. Unless stated to the contrary, all the experiments employ dust alkalinity derived from the average of the Journet et al. (2014) simulation as explained in Section 2.2.3

    Three initial run sets are conducted to establish reference results for comparison when coarse $NO_3^-$ formation is neglected; The noHC run assumes that no formation of $NO_3^-$, $NH_4^+$ or $SO_4^{2-}$ aerosol through heterogeneous chemistry on dust and SS, serving as a baseline to estimate the burden of gas condensing in particles in other configurations and the influence of
particle formation on nitrogen deposition rates. Additionally, two fTEQ runs are included to discuss the impact of ignoring the





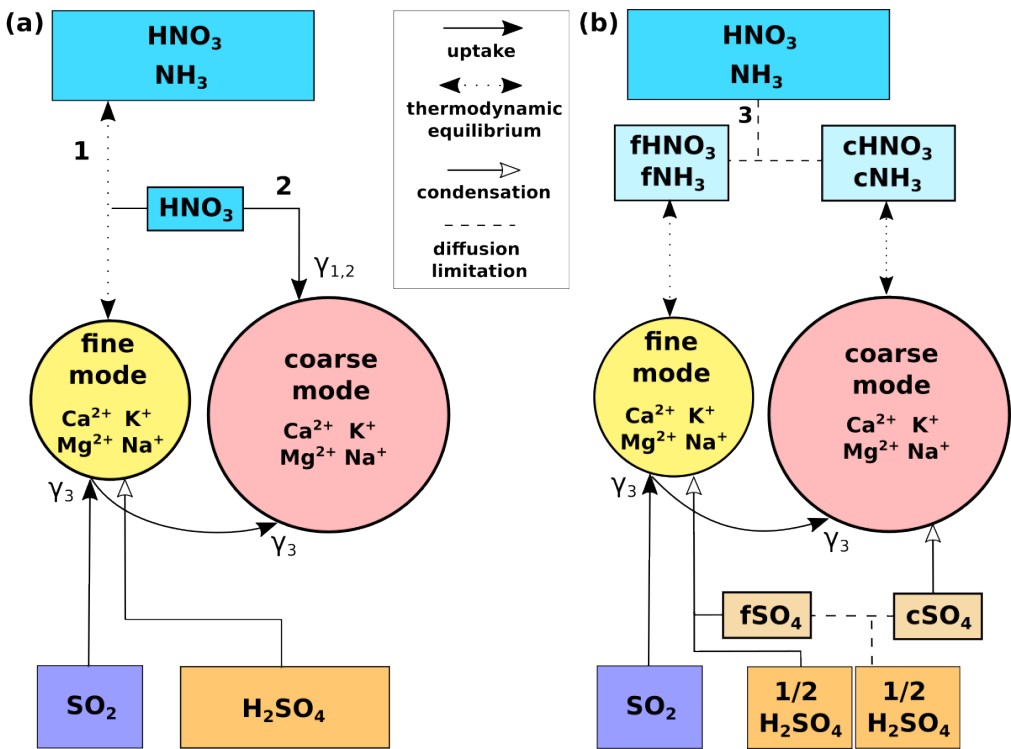

**Figure 2.** Illustration of the heterogeneous chemistry schemes of dust and SS developed in this work. (a) Schemes for the fTEQ and HYB mechanisms. (b) Scheme for DBCLL, that includes the kinetic diffusion limitation of gas species. The legend for line patterns is included on the center box. Numbers represent fTEQ (1), HYB (2) and DBCLL (3). In the bottom part, the pathways for processing $SO_{2(g)}$ and $H_2SO_{4(g)}$ gases for each scheme are represented; Namely, $SO_{2(g)}$ is uptaken by fine particles and the remaining gas is uptaken by coarse particles, while 50% of $H_2SO_{4(g)}$ is assumed to nucleate directly as fine $SO_4^{2-}$ and the rest is either condensed as fine $SO_4^{2-}$ or divided by diffusion limitation between fine and coarse modes, completely condensing on them. The uptake coefficients used for each uptake process (solid arrows) are indicated as $\gamma_1$ for $HNO_{3(g)}$ uptake on dust, $\gamma_2$ for $HNO_{3(g)}$ uptake on SS, and $\gamma_3$ for $SO_{2(g)}$ uptake on dust.

partitioning of semivolatile inorganic species on coarse particles. Specifically, fTEQ_noAlk neglects the presence of dust or SS in the aerosol mixture (R6-7 in Table 1), while fTEQ_du-ssAlk considers TEQ between gas and NVC in the fine modes of dust and SS particles (R6-12 in Table 1).

Next, we addressed the condensation of nitrate across the entire particle size range with runs employing the HYB and
DBCLL mechanisms. The HYB approach simplifies nitrate condensation on the coarse mode assuming an irreversible $HNO_{3(g)}$ UPTK reaction. We conducted two runs to explore the sensitivity of implementing the $HNO_{3(g)}$ UPTK reaction on coarse dust only (HYB_duUPTK) and on both coarse dust and SS (HYB_du-ssUPTK), comparing their results to assess the relative contribution of SS in heterogeneous chemistry under the assumption of $HNO_{3(g)}$ irreversible UPTK on the coarse mode.

In the HYB approach, all $HNO_{3(g)}$ and $NH_{3(g)}$ concentrations are initially available to condense in the fine mode through
TEQ reactions. Only the remaining $HNO_{3(g)}$ after TEQ calculation is considered for UPTK reactions in the coarse mode.

...
...




| Experiment | Fine TEQ | Coarse TEQ | Coarse DU UPTK | Coarse SS UPTK | DIFFLIM | DU Alk. | SS Alk. |
|---|---|---|---|---|---|---|---|
| noHC | | | | | | | |
| fTEQ_noAlk | ✓ | | | | | | |
| fTEQ_du-ssAlk | ✓ | | | | | J | ✓ |
| HYB_duUPTK | ✓ | | ✓ | | | J | |
| HYB_du-ssUPTK | ✓ | | ✓ | ✓ | | J | ✓ |
| HYB_DL | ✓ | | ✓ | ✓ | ✓ | J | ✓ |
| HYB_g0p1 | ✓ | | ✓ | ✓ | | J | ✓ |
| DBCLL_noAlk | ✓ | ✓ | | | ✓ | | |
| DBCLL_duAlk | ✓ | ✓ | | | ✓ | J | |
| DBCLL_du-ssAlk | ✓ | ✓ | | | ✓ | J | ✓ |
| DBCLL_ClaqAlk | ✓ | ✓ | | | ✓ | C | |

**Table 3.** The sensitivity experiments conducted in this study and their processes. *Fine TEQ* and *coarse TEQ* refer to the respective calculation of the TEQ on the fine and coarse modes of dust and SS, referred as *DU* and *SS*, respectively. *Coarse DU UPTK* and *Coarse SS UPTK* indicate the irreversible UPTK of $HNO_{3(g)}$ on coarse dust and SS, respectively. Alkalinity is denoted as *Alk.* and for the dust alkalinity column, alkalinity derived from the Journet et al. (2014) and Claquin et al. (1999) simulations is indicated by *J* and *C*, respectively. The experiments are grouped by sets of runs as described in Section 2.4.

This assumption may potentially lead to a misrepresentation of fine and coarse $NO_3^-$ formation, such as the underproduction of fine and overproduction of coarse $NO_3^-$ (Feng and Penner, 2007; Hauglustaine et al., 2014; Bian et al., 2017; Jones et al., 2021). To address this potential limitation, an additional simulation (HYB_DL) was conducted using the DIFFLIM calculation to distribute the $HNO_{3(g)}$ and $NH_{3(g)}$ gas that is kinetically available for condensation in the fine mode through TEQ and the
$HNO_{3(g)}$ that can form coarse $NO_3^-$ through UPTK reactions.

    Furthermore, to assess the influence of the $HNO_{3(g)}$ UPTK coefficient on results, instead of using a RH function (see Section 2.2.1) we conducted a simulation setting $\gamma(HNO_3) = 0.1$ for dust (HYB_g0p1), following experimental findings (Fenter et al., 1995; Hanisch and Crowley, 2001, 2003) and various modelling studies (Dentener et al., 1996; Liao et al., 2003; Bauer et al., 2004; Hodzic et al., 2006; Bauer et al., 2007; Feng and Penner, 2007).

Finally, we conducted four simulations using the DBCLL mechanism, which accounts for reversible heterogeneous nitrate chemistry both on fine and coarse modes. These simulations evaluate the influence of alkalinity, including the DBCLL_noAlk run excluding both dust and SS NVC content, the DBCLL_duAlk run accounting only for dust alkalinity, and the DBCLL_du-ssAlk run accounting for both dust and SS alkalinity. Additionally, since all three cases use dust alkalinity from the average of the Journet et al. (2014) simulation, an extra simulation was performed using dust alkalinity from the average Claquin et al.
(1999) simulations (DBCLL_Claq), instead of Journet et al. (2014), to assess the effect of the specific dust alkalinity content used.

    The model simulations were conducted on a global domain at a spatial resolution of $1.4$ degrees longitude by $1.0$ degree latitude, utilizing 48 hybrid pressure-sigma vertical layers up to 5hPa. The dynamics timestep was set to 180s and results were stored every 6 hours. The analysis period is the year 2018 after a spin-up period of half a year to initialize the concentration



fields. Meteorological variables were initialized from the ERA5 reanalysis (Hersbach et al., 2023) every 24h to keep the modeled circulation close to observations. A spin-up of 12h was used in each daily cycle before solving the chemistry. The initial state of the chemistry fields are those prognostically calculated by MONARCH the day before.

In addition to the meteorology-driven online emissions described in Sect. 2.1, the High-Elective Resolution Modelling Emission System version 3 (HERMESv3; Guevara et al., 2019) was employed to process both anthropogenic and biomass burning primary emissions. The global inventory CAMS-GLOB-ANT_v4.2 (Soulie et al., 2024) for 2016 was used for anthropogenic sources with updated temporal profiles that provide gridded monthly, day-of-the-year, day-of-the-week and hourly weight factors for the temporal disaggregation of emitted fluxes (Guevara et al., 2021). The biomass burning emissions are provided by the GFASv1.2 dataset (Kaiser et al., 2012), which accounts for forest, grassland, and agricultural waste fires derived from satellite products. Oceanic natural emissions of DMS are provided by CAMS-GLOB-OCE v3.1 (Lana et al., 2011; Denier van der Gon et al., 2023).

Table S1 summarizes the total emissions (anthropogenic, biogenic and biomass burning) used in this work. The emitted mass of the main anthropogenic aerosol precursors are 104.4 Tg for $SO_{2(g)}$, 93.8 Tg for $NO_{x(g)}$, and 61.8 Tg for $NH_{3(g)}$.

## 2.5 Model evaluation

The model results are evaluated against surface observational datasets from several networks sourced from the Globally Harmonised Observational Surface Treatment (GHOST) project, an initiative of the BSC's Earth Department dedicated to the harmonisation of publicly available global surface observations (Bowdalo et al., 2024). Supplementary Figure S13 shows the stations used for each of the species analyzed. For gas and aerosol nitrate, ammonia and sulfate species, GHOST include datasets from the Clean Air Status and Trends Network (US-EPA-CASTNET), the US EPA Air Quality System (US-EPA-AQS) and the Canada National Air Pollution Surveillance Program (NAPS) for North and Central America, the East Asia Acid Deposition Monitoring Network (EANET) for Asia, and the EBAS and the European Environmental Agency Air Quality (EEA AQ eReporting) for Europe.

These observations are filtered for rural and background sites only, in order to exclude stations near emission sources not representative of the background conditions depicted by the model resolution, which describes the long-range transport of $NO_3^-$, $NH_4^+$ and $SO_4^{2-}$ formation. However, information on the station type was not available for the EANET and US-EPA-CASTNET data. For these networks, all stations were used, a factor that has to be accounted for when evaluating the results.

Regarding $PM_{2.5}$ and $PM_{10}$, GHOST includes data from the CHILE-SINCA network for Chile, the Beijing Municipal Ecological and Environmental Monitoring Center (BJMEMC), the China National Environmental Monitoring Centre (CNEMC), WMO World Data Center for Aerosols (EBAS-WMO-WDCA) for Europe, the Japan National Institute for Environmental Studies (NIES), the Ministerio de Transición Ecológica (MITECO) for Spain, the UK AIR network for United Kingdom, and the global AirNow DOS network (US-EPA-AirNow-DOS). For these networks, the station type was not included as a criteria to include it in the evaluation. The statistical metrics used in the evaluation are outlined in the Supplementary Section S3, and the quality flags for the selected stations in Table S8.





Additionally, we compare our model results with the budgets reported in the literature (Hauglustaine et al., 2014; Bian et al., 2017; Rémy et al., 2022). Namely, the AeroCom phase III nitrate experiment (Bian et al., 2017) intercompares global

$NO_3^-$ budgets from nine global models for the year 2008. Particular discussion is devoted to results shown there for the GMI Bian et al. (2009) and EMAC (Karydis et al., 2016) models, that introduce relevant approaches of interest to our work. Complementary, results from Hauglustaine et al. (2014) and Rémy et al. (2022) are also used. The comparison provides a qualitative view of current estimates of particulate $NO_3^-$ formation in the atmosphere and the role of representing key processes in models. Since results from the literature are provided for different years, some of the differences may be attributed to changes

in emissions and environmental conditions.

## 3  Results and discussion

### 3.1  Spatial distributions

The global spatial distribution of nitrate species varies significantly based on the formation mechanisms assumed. We present the results by incrementally increasing complexity to assess the impact of each specific formation process on the global nitrate

distribution. Initially, we neglect the formation of coarse $NO_3^-$, focusing solely on nitrate formation in the fine mode. This is driven by pure nitrate-ammonia-sulfate neutralization (reactions R6-7 in Table 1), both with and without considering the effect of fine dust and SS alkalinity (reactions R8-12 in Table 1). We then discuss the formation of coarse $NO_3^-$ on dust and SS particles, considering both irreversible uptake processes (reactions R4-5) and reversible condensation processes (reactions R8-R12). Spatial distributions of surface concentrations, column loads and zonal average concentrations are presented for

$HNO_{3(g)}$ and particulate $NO_3^-$ (Figure 3, Figure S1, and Figure S2), $NH_{3(g)}$ and $NH_4^+$ (Figure S4, Figure S5, and Figure S6), and $SO_{2(g)}$ and $SO_4^{2-}$ (Figure S7, Figure S8, and Figure S9). These results from the different sensitivity runs (see Table 3) are averaged for 2018.

### 3.1.1  Effects of omitting coarse nitrate formation

The fTEQ_noAlk sensitivity run employs the TEQ model to form $NO_3^-$ exclusively in the fine mode (fTEQ). In this scenario,

$HNO_{3(g)}$ and $NH_{3(g)}$ condensation is calculated without considering dust and SS alkalinity in the bulk aerosol phase, focusing only on pure nitrate-ammonia-sulfate neutralization (reactions R6-7 in Table 1). Fine particulate $NO_3^-$ primarily forms near regions with significant anthropologic pollution, where emissions of $NO_{x(g)}$, $NH_{3(g)}$, and $SO_{2(g)}$ are dominant, such as Northern China, India, Europe and eastern North America, with concentrations ranging from 2 to 10 $\mu g\ m^{-3}$ (Figure 3e). At the surface level $NH_4^+$ is mostly associated with fine $NO_3^-$, presenting concentrations of 1-3 $\mu g\ m^{-3}$ (Figure 3e and Figure S4e).

At higher altitudes, fine $NH_4^+$ also forms in the presence of dust and SS, with column burdens ranging from 1 to 3 $mg\ m^{-2}$ (Figure S5 and Figure S8). This is due to $NH_4^+$ neutralizing particulate $SO_4^{2-}$, which forms from the condensation of $H_2SO_{4(g)}$ and $SO_{2(g)}$ on fine dust and SS. The omission of dust and SS alkalinity results in very low values of aerosol pH globally; Over





oceanic and dusty regions, pH values range from 1 to 2, while in industrialized regions such as Europe and Asia, pH can reach up to 5 (Figure 4a).

Including dust and SS NVC neutralization, along with the nitrate-ammonia-sulfate neutralization process (reactions R8-12 of Table 1), significantly increases arosol pH in the fTEQ_du-ssAlk run (Figure 4b). This effect is especially pronounced over the open ocean, where pH increases from 1 to 5, and over dust source regions such as the Sahara desert, where pH increases from 1 to 9. The elevated pH notably affects the partitioning of $HNO_{3(g)}$, reducing its column burdens throughout the Northern Hemisphere by approximately 5 $mg\ m^{-2}$ compared to the fTEQ_noAlk scenario (Figure S1d, g).

The fTEQ_du-ssAlk run has minimal impact on $HNO_{3(g)}$ surface concentrations over land (Figure 3d, g). The concentrations decrease only in the upper troposphere (above 500hPa, Figure S2d, g) by about 0.5 $\mu g\ m^{-3}$. However, the column burdens of fine $NO_3^-$ increase significantly, by more than 5 $mg\ m^{-2}$ over dusty regions such as the Saharan desert and Middle East (Figure S1e, h). This suggests that the condensation of $HNO_{3(g)}$ on dust and SS during long-range transport is a key driver of nitrate formation in this scenario. In remote oceanic areas, fine $NO_3^-$ column burdens increase by approximately 1 $mg\ m^{-2}$,

with transoceanic transport at low latitudes showing column loads ranging from 4 to 7 $mg\ m^{-2}$ (Figure S1h). At the surface, fine $NO_3^-$ concentrations of 0.5 $\mu g\ m^{-3}$ are present across the equatorial belt, with higher values over remote dusty regions (up to 2 $\mu g\ m^{-3}$), though no significant increase is observed over polluted areas such as Europe and Asia. These results indicate that fine $NO_3^-$ form predominantly on dust rather than on SS, attributed to the relatively higher alkalinity of dust compared to SS, as derived from Journet et al. (2014) dataset (see Sect. 2.2.3). Overall, incorporating reactions R8-12 increases fine $NO_3^-$

concentrations by approximately 0.5 $\mu g\ m^{-3}$ beyond polluted areas at ground level and across the equatorial belt (Figure 3e, h and Figure S2e, h). This increase is noteworthy and highlights the sensitivity of nitrate formation to the alkalinity of dust and SS, as it will be further demonstrated in the surface observational evaluation in Section 3.2.

### 3.1.2  Effects of assuming irreversible formation of coarse nitrate

A commonly employed strategy in atmospheric chemistry models to represent nitrate formation on coarse particles in global

simulations is the incorporation of irreversible heterogeneous reactions. In this study, we examine the results of the HYB sensitivity runs (detailed in the second block of Table 3, where fine $NO_3^-$ formation is simulated using fTEQ (reactions R6-12 in Table 1), while different irreversible uptake reactions (and rates) are explored to model coarse $NO_3^-$ (UPTK reactions R4 and R5 in Table 1). The HYB method excludes the possibility of particulate coarse $NO_3^-$ evaporation, potentially leading to positive biases in its global burdens.

The sensitivity run HYB_duUPTK focuses on coarse $NO_3^-$ formation exclusively on dust, utilizing reaction R4 while disabling any process related with SS particles (R5 disabled). This results in the redistribution of available $HNO_{3(g)}$ towards the production of coarse $NO_3^-$ in regions predominantly affected by dust, such as the Middle East and East Asia, where column loads range on average between 0.5-4.0 $mg\ m^{-2}$ across the Northern Hemisphere, peaking at 10-20 $mg\ m^{-2}$ (Figure S1 l). Notably, this impact is less pronounced in the Saharan dust belt, where the availability of $HNO_{3(g)}$ for coarse $NO_3^-$ formation

is limited. The production of coarse $NO_3^-$ coincides with a modest reduction in fine $NO_3^-$ column loads over dusty regions (around 1 $mg\ m^{-2}$) compared to the fTEQ_du-ssAlk simulation, attributed to diminished availability of $HNO_{3(g)}$ for fTEQ





reactions. Surface concentrations of coarse $NO_3^-$ range from 2 to 10 $\mu g\ m^{-3}$ over China, India, and the Middle East (Figure 3 l). Long-range transport of $NO_3^-$ is significantly influenced by dust NVC loads, showcasing notable transatlantic and transpacific transport of coarse $NO_3^-$ (Figure S1). This transport predominantly occurs below 800hPa, with concentrations around

$1\mu g\ m^{-3}$, while lower concentrations of coarse $NO_3^-$ are also discernible at altitudes up to 400hPa (Figure S2).

The role of SS is investigated by enabling both R4 and R5 in the HYB_du-ssUPTK run. In reaction R5, a constant uptake coefficient of $HNO_{3(g)}$ on coarse SS ($\gamma_{SS}$) of 0.05 is used, while in R4, the uptake coefficient on coarse dust ($\gamma_{dust}$) is parameterized following Fairlie et al. (2010) as a function of RH with an average value of $\gamma_{dust} = 5.3 \cdot 10^{-4}$, as discussed in section 2.2.1. The uptake of $HNO_{3(g)}$ on SS enhances the formation of coarse $NO_3^-$ over open ocean and coastal areas

affected by SS outbreaks (Figure S11). Notably, surface concentrations in Europe and North-America reach 1-2 $\mu g\ m^{-3}$, with regions exceeding 5 $\mu g\ m^{-3}$ significantly extended (Figure 3o). Coarse $NO_3^-$ becomes also notable over West Siberia, East South-America and South Africa (0.5-1 $\mu g\ m^{-3}$) and over remote oceanic regions (0.2 $\mu g\ m^{-3}$), with its presence extended to high altitudes in the Southern Hemisphere (Figure S2o). The long-range transport of coarse $NO_3^-$ is enhanced, particularly over oceans, compared to the HYB_duUPTK case, by approximately 3.5 $mg\ m^{-2}$ (Figure S1o). This increase is attributed

to a higher depletion of $HNO_{3(g)}$ by irreversible reactions R4 and R5 (Figure S1m). Consequently, the HYB_du-ssUPTK sensitivity run exhibits lower $HNO_{3(g)}$ concentrations in the atmosphere compared to the other simulations.

As previously noted, the $\gamma_{dust}$ values used in both HYB_duUPTK and HYB_du-ssUPTK runs are well below the value used for $\gamma_{SS}$ and other studies in the literature (Hauglustaine et al., 2014; Rémy et al., 2022). To assess the sensitivity to more efficient R4 production rates, the HYB_g0p1 run employs a constant value for $\gamma_{dust}$ of 0.1, rather than utilizing the RH-

dependent function proposed by (Fairlie et al., 2010). Results indicate an increase in coarse $NO_3^-$ formation in regions rich in dust (Figure S10), such as the Sahara desert, the Persian Gulf and East Asia (Figure S3c). Furthermore, a slight enhancement in the transport of coarse $NO_3^-$ across northern latitudes is observed, despite lower dust concentrations in that region. This can be attributed to the predominant production of coarse $NO_3^-$ on dust across continental Eurasia, followed by its long-range transport at high altitudes to North America. Although rising the value of $\gamma(dust)$ leads to a significant increase compared to

HYB_du-ssUPTK, the changes are not excessively substantial, suggesting that the availability of $HNO_{3(g)}$ may be the limiting factor in R4. Furthermore, the HYB mechanisms preferentially condense $HNO_{3(g)}$ via TEQ in the fine mode and over SS through R5 in the coarse mode, rather than over coarse dust through R4, as noted here and also assessed in next sections.

A possible explanation for the excessive condensation of $HNO_{3(g)}$ over dust and SS in the HYB mechanism might be the

unrestricted availability of $HNO_{3(g)}$ for $NO_3^-$ formation. To address this potential issue, we performed a sensitivity test using the DIFFLIM calculation on the HYB_du-ssUPTK run (HYB_DL). This approach limits $HNO_{3(g)}$ availability for fine TEQ and coarse UPTK based on the particulate surface for each mode. The DIFFLIM methodology redistributes $NO_3^-$ column and surface concentrations from the fine to the coarse mode, resulting in coarse $NO_3^-$ column burdens slightly below those obtained in HYB_g0p1. Coarse $NO_3^-$ formation rises particularly over the Sahara and across the Atlantic and Pacific oceans. This can

be explained by HYB_DL preserving more $HNO_{3(g)}$ for reaction via R4 and R5 in the coarse mode, thereby enhancing its condensation over coarse dust.



Comparing the spatial distributions obtained by models EMEP, INCA and GMI in Bian et al. (2017), that use a similar HYB approach to our HYB_du-ssUPTK run in MONARCH, reveals similar trends. Both studies show significant formation of total $NO_3^-$ (fine + coarse) over polluted regions, although our results generally exceed the column loads reported by AeroCom models, ranging from 7 to 16 $mg\ m^{-2}$ compared to 10-25 $mg\ m^{-2}$ (Figure S1n, o). AeroCom models do not show such a prominent transport of coarse $NO_3^-$ across the North Atlantic as observed in our study (0.2-0.5 $mg\ m^{-2}$ vs. 4 $mg\ m^{-2}$, respectively). In Section 3.4 we delve into the excessive formation of particulate $NO_3^-$ formation in the HYB mechanism in terms of the total nitrogen budget.

A closer comparison with Hauglustaine et al. (2014), using the LMDz-INCA model with similar $HNO_{3(g)}$ UPTK coefficients on dust and SS as in MONARCH (also utilizing a function of RH as in Fairlie et al. (2010)), shows strong agreement, particularly with our HYB_duUPTK run. For instance, concentrations of total $NO_3^-$ over polluted areas are similar in both cases in terms of geographical distribution and magnitude (14-20 $mg\ m^{-2}$ in HYB_duUPTK compared to 10-20 $mg\ m^{-2}$ in Hauglustaine et al. (2014), Figure S1 l). Hauglustaine et al. (2014) also observes fine $NO_3^-$ transport downwind of the Sahara and coarse $NO_3^-$ across the northern Atlantic, with column burdens around 1-2 $mg\ m^{-2}$, slightly below our results of 1-5 $mg\ m^{-2}$ for HYB_duUPTK. This resemblance of Hauglustaine et al. (2014) with the HYB_duUPTK - and not with HYB_du-ssUPTK run - can be attributed to the $HNO_{3(g)}$ UPTK coefficient for SS employed by Hauglustaine et al. (2014): Instead of a constant value as implemented in HYB_duUPTK, it uses a function of RH ranging from $1 \cdot 10^{-3}$ to 0.1, therefore forming less $NO_3^-$ over SS than our HYB_du-ssUPTK run. This comparison with Hauglustaine et al. (2014) might point at a potential overestimation of the UPTK coefficient of $HNO_{3(g)}$ on SS in the HYB_du-ssUPTK simulation, conclusion that is further discussed in the observational evaluation (Section 3.2) and in the budgets analysis (Section 3.4). Additionally, while Hauglustaine et al. (2014) attributes transatlantic transport of particulate $NO_3^-$ to the UPTK of $HNO_{3(g)}$ on SS, our study relates north Atlantic transport of coarse $NO_3^-$ with $HNO_{3(g)}$ UPTK on dust rather than on SS, as indicated by its increased presence in the HYB_g0p1 simulation. As a final remark, it is important to take into account that, as Hauglustaine et al. (2014), also HYB_duUPTK overestimates particulate $NO_3^-$ when compared to observations, as explained in Section 3.2. To further elucidate the source of the HYB_du-ssUPTK overestimation of coarse $NO_3^-$, we compare our spatial distributions with those from Jones et al. (2021), which employs a similar HYB mechanism using the Met Office Unite Model (UM). Jones et al. (2021) computes fine $NO_3^-$ formation with adaptations from Hauglustaine et al. (2014) (by testing different accommodation coefficients for $HNO_{3(g)}$ onto preexisting ammonium-nitrate aerosols, although reporting almost negligible effects on coarse $NO_3^-$ concentrations) and forms coarse $NO_3^-$ from the $HNO_{3(g)}$ UPTK using coefficients from Fairlie et al. (2010) for dust and Burkholder et al. (2020) for SS. Consequently, its coarse $NO_3^-$ distribution is comparable to that obtained from the HYB_du-ssUPTK run, despite using different coefficients for the $HNO_{3(g)}$ UPTK on SS. Some differences arise between both models: HYB_du-ssUPTK reports significantly lower sub-Saharan concentrations of coarse $NO_3^-$, while showing higher formation of coarse $NO_3^-$ over polluted areas such as Europe, east North-America and Asia (4 vs. 8-20 $mg\ m^{-2}$), as well as over oceans (0.4-2.0 vs. 1.5-3.0 $mg\ m^{-2}$). Remarkably, the North-Atlantic transport of coarse $NO_3^-$ is only present in our study with 2.0-4.0 $mg\ m^{-2}$. Their results, when compared with the HYB_duUPTK simulation, also show lower coarse $NO_3^-$ concentrations over North-America (2-3 vs. 1-2 $mg\ m^{-2}$).





Furthermore, a global budget comparison between HYB_duUPTK and Jones et al. (2021) indicates similar high coarse $NO_3^-$ formation in both models. This suggests a possible excessive formation of coarse $NO_3^-$ in HYB_duUPTK, as both models exhibit similar distributions of coarse $NO_3^-$, albeit Jones et al. (2021) also includes SS in its formation. However,

several alternative explanations for the differences between both models could be considered, including variations in the fine $NO_3^-$ formation mechanism, differences in the study period (2018 vs. 20-year period), and differences in the parameterization of $HNO_{3(g)}$ UPTK on SS. Detailed budget comparisons are presented in Section 3.4.

### 3.1.3 Effects of accounting for reversible formation of coarse nitrate

Advancing the representation of dust and SS heterogeneous chemistry in models involves incorporating the partitioning of

semivolatile inorganic species as a fully reversible process across all particle sizes. To explore the implications of this advancement, we conducted sensitivity runs using the DBCLL mechanism, as introduced in Section 2.3. The DBCLL runs (outlined in Table 3) are designed to assess the impact of the reversible chemistry and sensitivities associated to dust and SS NVC participation in the partitioning.

Firstly, in the DBCLL_noAlk sensitivity run, $NO_3^-$ formation is exclusively considered through reactions R6-7 (see Table

1). Here, the mass of dust and SS serves solely to determine the available particle surface area in the DIFFLIM calculation, with alkalinity not factored into the partitioning assessment. Notably, the spatial distributions of $HNO_{3(g)}$ and fine $NO_3^-$ in the DBCLL_noAlk run closely resemble those in the fTEQ_noAlk run. $HNO_{3(g)}$ remains predominantly in the gas phase across the Northern Hemisphere (Figure S1p). Fine $NO_3^-$ primarily associates with fine $NH_4^+$ in regions without dust or SS, such as China, India, Europe and east North-America (Figure 3q), with minimal concentrations of $NO_3^-$ simulated at higher

altitudes (Figure S2q). Conversely, fine $NH_4^+$ is present at elevated altitudes alongside sulfate particles, similar to the findings in the fTEQ_du-ssAlk case. In contrast, coarse $NO_3^-$ is nearly absent in the DBCLL_noAlk run, with minimal concentrations observed at the surface level over northern China at 0.4 $\mu g\ m^{-3}$, showing no apparent correlation with the presence of dust and SS (Figure 3r). Similarly, the formation of coarse $NH_4^+$ is very limited, with some presence over the Persian Gulf and the Sahara (0.02-0.1 $mg\ m^{-2}$ column loads), primarily in anthropologically polluted areas, where it forms alongside coarse $SO_4^{2-}$

at 0.2 $mg\ m^{-2}$ (Figure S5r and Figure S8r). The simultaneous formation of coarse $NH_4^+$ and $SO_4^{2-}$ particles can be attributed to diffusion limitation, which allocates $NH_{3(g)}$ to neutralize sulfate particles present in the coarse mode.

The differences between fTEQ_noAlk and DBCLL_noAlk illustrate the impact of assuming the diffusion limitation before the partitioning of semi-volatile species. While fine $NO_3^-$ formation remains consistent between both runs, allowing all acid species to be available for partitioning into the fine mode in fTEQ_noAlk results in enhanced fine particle production and

long-range transport, which is not observed in DBCLL_noAlk. Consistently, the pH in the fine mode closely mirrors that of fTEQ_noAlk, while the pH of particles in the coarse mode indicates even greater acidity (Figure 4f1, f2)).

The impact of dust NVC on the DBCLL mechanism is introduced through reactions R8-11 (Table 1) in the DBCLL_duAlk sensitivity run. Compared with DBCLL_noAlk, dust alkalinity significantly increases pH values (i.e. more basic) over the Sahara and continental Asia to a range of 6 to 8 (Figure 4g1, g2), thereby enhancing the formation of both fine and coarse $NO_3^-$

as well as coarse $SO_4^{2-}$ in these regions. Additionally, fine $NO_3^-$ formation is amplified in polluted areas, with surface con-



centrations rising beyond 5 $\mu g\ m^{-3}$ (an increase of 2-5 $\mu g\ m^{-3}$ relative to DBCLL_noAlk). This enhanced formation extends to regions rich in coarse dust, such as the Persian Gulf (surface concentrations of 1-3 $\mu g\ m^{-3}$) and central/downwind areas of the Saharan desert (surface concentrations of 0.2-0.5 $\mu g\ m^{-3}$ and loads of 2.0 $mg\ m^{-2}$), as shown in Figures 3u and S1u. Over these areas, DBCLL_duAlk nearly depletes $HNO_{3(g)}$ gas (remaining surface concentrations of 0.01 to 0.05 $\mu g\ m^{-3}$).

At high altitudes, both fine and coarse $NO_3^-$ are distributed across the Atlantic and Pacific oceans with column loads of 1-2 $mg\ m^{-2}$ (Figure S1u), resembling the transoceanic patterns also observed in the HYB mechanisms. Conversely, compared to the DBCLL_noAlk run, fine $NH_4^+$ increases over Europe and North-Central America (by at least 1 $mg\ m^{-2}$, discussed in Section 3.3.1), while it decreases over dusty areas (by 1.0 to 2.0 $mg\ m^{-2}$, Figure S5t, q). This pattern is consistent with the comparison between the fTEQ_noAlk and the fTEQ_du-ssAlk runs (Figure S5h, e). Concurrently, coarse $NH_4^+$ formation

increases significantly in transoceanic areas, reaching 0.08-0.1 $mg\ m^{-2}$ alongside coarse $NO_3^-$ particles, although its formation is halted over dusty regions (i.e. Middle East and Sahara, Figure S5u). This inhibition can be attributed to the increased alkalinity of dust, which neutralizes $HNO_{3(g)}$ and limits the condensation of $NH_{3(g)}$. Lastly, the introduction of dust alkalinity strongly enhances the formation of coarse $SO_4^{2-}$, both over polluted regions (5 $mg\ m^{-2}$) and remote areas (0.2-1.0 $mg\ m^{-2}$), as shown in Figure S8. The enhanced production of coarse particles (principally coarse $NO_3^-$) provides additional surface area

for $H_2SO_{4(g)}$ to condense on.

We also incorporated the effect of SS NVC into the nitrate formation adding reaction R12 (Table 1) in the DBCLL_du-ssAlk run. Considering both dust and SS alkalinity produces an important increase in oceanic pH, from 1 to 5-6 in the fine mode and above 7 in the coarse mode (Figure 4h1, h2) and redistributes the $NO_3^-$ partitioning towards the coarse mode. Compared to DBCLL_du-Alk (which accounts only for dust alkalinity), fine $NO_3^-$ concentrations areapproximately halved

across all regions, while coarse $NO_3^-$ surface concentrations and loads roughly double (Figure 3w, x and Figure S1w, x). Concentrations of coarse $NO_3^-$ are remarkable in the transatlantic and transpacific transport, with column loads of 2-4 $mg\ m^{-2}$, surface concentrations of 0.4 $\mu g\ m^{-3}$ and zonal average of 0.5-1.0 $\mu g\ m^{-3}$). Over continental areas with elevated coarse dust concentration, coarse $NO_3^-$ column loads reach 4-5 $mg\ m^{-2}$ and surface concentrations are 0.5-1.0 $\mu g\ m^{-3}$ . This increase is accompanied by a rise in $HNO_{3(g)}$ surface concentrations, from 0.1 to 1.0 $\mu g\ m^{-3}$ in the main polluted areas (Figure 3s,

v). The inclusion of both dust and SS alkalinity in the coarse TEQ calculation enhances coarse $NO_3^-$ formation directly over and downwind of dusty areas, compared to accounting for dust alkalinity alone (DBCLL_duAlk). This enhancement can be attributed by SS NVC - such as $Cl^-$ (see Section 2.2.3) - halting fine $NO_3^-$ formation, leaving more $HNO_{3(g)}$ in the gas phase to be transported over dusty regions, where it subsequently forms coarse $NO_3^-$.

The spatial distributions of coarse $NO_3^-$ in the DBCLL runs show a significant contrast compared to those obtained with the

HYB runs. Formation is concentrated directly over and downwind of dusty areas, rather than over anthropologically polluted areas as seen when using HYB mechanisms. For example, in the DBCLL_du-ssAlk, surface concentrations over Europe do not exceed 0.5 $\mu g\ m^{-3}$ (Figure 3x). Including SS alkalinity causes fine $NH_4^+$ loads to decrease over Europe and Eastern North-America from 3 to 1 $mg\ m^{-2}$, while slightly lower values are reported over Asia (at 2-3 $mg\ m^{-2}$), likely due to the reduced presence of fine $NO_3^-$. Conversely, coarse $NH_4^+$ formation is substantially enhanced over Europe and Eastern North-America

(0.1-0.2 $mg\ m^{-2}$), Asia (reaching 1.0 $mg\ m^{-2}$) and transoceanic regions (0.1 $mg\ m^{-2}$) (Figure S5x). Although the presence





of coarse $NH_4^+$ increases over the Sahara and the Middle East, this may be a consequence of long-range transport from polluted areas. The decrease in fine $NH_4^+$ and the increase in coarse $NH_4^+$ are directly related to the particulate $NO_3^-$ patterns, outlining the strong correlation between these species when TEQ is employed for both fine and coarse gas-aerosol partitioning. This emphasizes the important role of the basic NVC from SS particles (i.e. $Na^{2+}$), illustrating the sensitivity of the results to the inclusion of SS alkalinity.

Overall, the major sensitivity identified in the intercomparison of mechanisms is attributed to reactions R8 to R12, where the alkalinity of dust and SS modifies dramatically $NO_3^-$ spatial patterns. The changes observed across the different mechanisms analyzed are consistent and highly sensitive to the presence of NVC in the partitioning of semi-volatile species. The sole inclusion of the DIFFLIM calculation in the DBCLL mechanism has a minor impact, as shown by the comparison of the fTEQ_noAlk and DBCLL_noAlk runs. An additional simulation was conducted to explore the sensitivity of $NO_3^-$ formation specifically on dust alkalinity using a different mineralogical dataset. Specifically, we derived the average alkalinity of dust from Claquin et al. (1999) mineralogical dataset instead of Journet et al. (2014), DBCLL_ClaqAlk run (Table 3), as detailed in Section 2.2.3. The results of this run show significant differences in specific arid regions compared with DBCLL_du-ssAlk. Notably, $NO_3^-$ load increases substantially over the Middle East and northern India, while a slight reduction is observed over northern Africa. These changes are significant enough to impact the long-range transport of both fine and coarse $NO_3^-$ across the northern hemisphere (Figure S3k, l). Coarse $NO_3^-$ decreases by about 1-2 $mg\ m^{-2}$ over and downwind of dusty areas, while fine $NO_3^-$ increases by 2-5 $mg\ m^{-2}$ along the equatorial belt and over polluted regions. Similar differences are found in particulate $NH_4^+$, although with an order of magnitude lower. Notably, $HNO_{3(g)}$ distributions report negligible differences between both runs. The lower alkalinity derived from the Claquin et al. (1999) dataset accounts for these differences, primarily affecting the formation of $NO_3^-$ during long-range transport of dust. This is further discussed in the following section 3.4.

It is insightful to compare the results from the DBCLL_du-ssAlk run with those from the EMAC model (Karydis et al., 2016), which employs a similar configuration to DBCLL, but first performing the TEQ for the bulk gas and aerosol mass, and subsequently redistributing the concentrations with the DIFFLIM calculation. Additionally, EMAC makes use of globally heterogeneous dust alkalinity based on 12 mineralogy source data points. Results from the EMAC model are reported in Bian et al. (2017) and Karydis et al. (2016) (Table 5 and Supplementary Section S4). Surface distributions of aerosol $NO_3^-$ are closely aligned in both models over Europe and North America, with concentrations ranging from 1 to $3\mu g\ m^{-3}$. However, DBCLL_du-ssAlk tends to present higher $NO_3^-$ concentrations over Asia, India and Middle East compared to the EMAC averages for those regions (5-10$\mu g\ m^{-3}$ vs. 6-15$\mu g\ m^{-3}$, respectively). Additionally, our results show some variations over secondary areas compared to Karydis et al. (2016). For instance, Karydis et al. (2016) reports various deviations to observations, like coarse $NO_3^-$ underestimation over continental regions such as southern Europe, central-east Asia, Middle East, and south-west USA. In the study, these deviations to observations are attributed to high concentrations of $H_2SO_4^{2-}$ condensed on dust particles over those areas, a problem mitigated in our study since it presents lower sulfate concentrations over these regions (Figure S7). Another bias with observations noted in Karydis et al. (2016) is the elevated fine $NO_3^-$ formation over the Arctic, a phenomenon not observed in our case, likely due to the restriction of RH and temperature when using ISORROPIA-II for TEQ (see Section 2.2.2). Additionally, Karydis et al. (2016) noted that their results overestimated coarse $NO_3^-$ in central Africa (1-2



$\mu g\ m^{-3}$ surface concentrations) due to excessive $HNO_{3(g)}$ from biomass burning coupled with low $H_2SO_4^{2-}$ concentrations. They suggested that an HYB approach could solve this bias. In our case we report much lower values of coarse $NO_3^-$ over central Africa, both with the DBCLL_du-ssAlk (0.3-1.0 $\mu g\ m^{-3}$) and with the HYB mechanisms (0.3-0.8 $\mu g\ m^{-3}$). However, since we do not register high $HNO_{3(g)}$ concentrations over that region in the baseline simulation noHC (with 0.04-0.1 $\mu g\ m^{-3}$,
Figure 3a) compared to Karydis et al. (2016), it indicates that we might be overestimating coarse $NO_3^-$ over central Africa with both DBCLL and HYB mechanisms. This implies that the HYB mechanism might not be a solution to improve coarse $NO_3^-$ predictions over that area. In this context, total particulate $SO_4^{2-}$ and $NH_4^+$ are in close agreement in both EMAC and the DBCLL_du-ssAlk run (Figures S4w, x and S7w, x), with only slightly higher formation of coarse $NH_4^+$ over the India peninsula in MONARCH compared to EMAC (2.5-3.0 $\mu g\ m^{-3}$ vs. 0.9-2.0 $\mu g\ m^{-3}$, respectively).

## 3.2 Evaluation with observations

We evaluate each sensitivity run by comparing with measured surface concentrations of relevant species involved in nitrate formation, as detailed in Section 2.5. This evaluation includes an analysis of statistics and time series of monthly mean values, focusing on the ability of the various mechanisms under study to capture observed nitrate variability across the globe. The globally averaged results are shown in Figure 5, while regionalized results for Europe, Asia, and Central-North America are
presented in Figure S12. It is important to note that the number of stations and spatial coverage vary significantly depending on the type of species, as illustrated in Figure S13. Consequently, certain evaluation metrics may not be representative of the model's performance globally, and might only reflect accuracy within specific regions. The correlation coefficients, bias and RMSE metrics (Section S3) for the different sensitivity simulations are reported in Table 4.

### 3.2.1 Nitrate species

Fine particulate nitrate ($PM_{2.5}NO_3$) is formed in the same way among mechanisms (through TEQ), with the exception of DB-CLL that implements a DIFFLIM coefficient to limit its formation. The fTEQ and HYB mechanisms present higher $PM_{2.5}NO_3$ from October to April (0.5-1.0 $\mu g\ m^{-3}$) than DBCLL, which reports approximately half of the concentration. This gap is more pronounced over North and Central America than over Europe, although both present a similar pattern. Differences among mechanisms are reduced from April to October (summer and fall in northern hemisphere), with the exception of HYB_du-
ssUPTK that shows artefacts, significantly overestimating $PM_{2.5}NO_3$ during these months. Notably, the runs considering the effect of dust and SS alkalinity in the partitioning process reduce the negative bias of $PM_{2.5}NO_3$ significantly, especially during February. Increased variability among experiments is found in the evaluation of $HNO_{3(g)}$ and Total $NO_3^-$ (with more observational data available over North-Central America compared to Europe and Asia). This variability is highly influenced by the pathway used for coarse $NO_3^-$ formation in each mechanism. Specifically, larger variability is noted over Asia compared
to Europe and North-Central America (Figure S12).

From the various mechanisms analysed, the representation of coarse $NO_3^-$ formation through an irreversible uptake reaction (HYB runs) consistently yield overestimated surface concentrations of Total $NO_3^-$ in all the regions analyzed. Better agreement is found under the assumption of reversible partitioning (DBCLL).





**Figure 3.** Surface concentration ($\mu g\ m^{-3}$) of $HNO_{3(g)}$, fine and coarse $NO_3^-$ simulated by the different mechanisms, averaged for 2018.





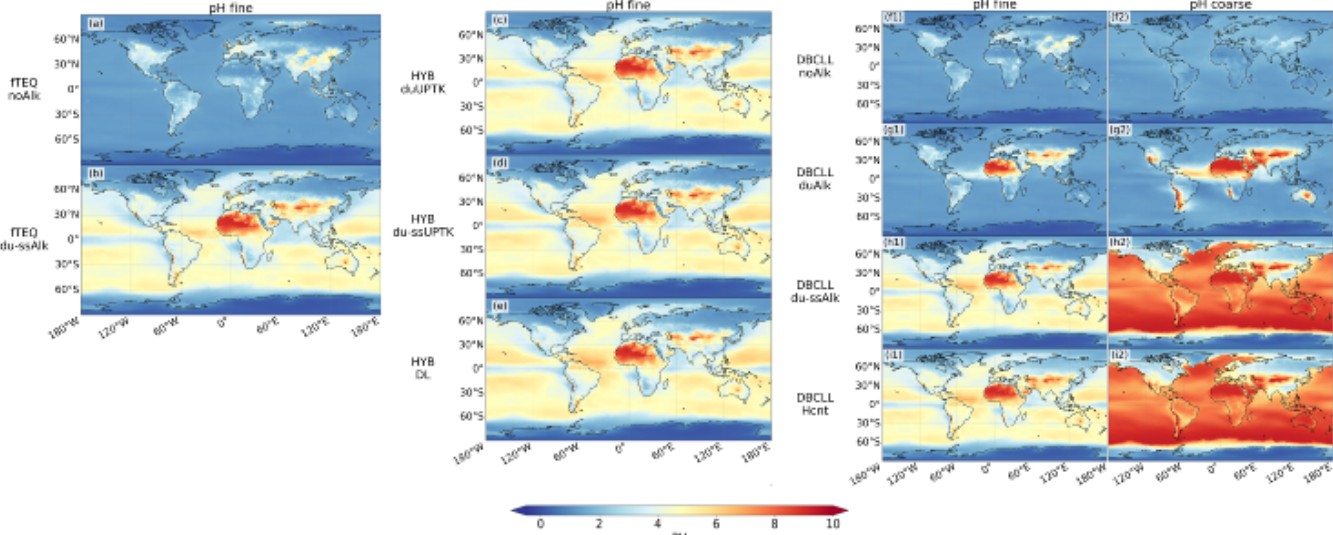

**Figure 4.** Surface average pH of the studied mechanisms for fine and coarse aerosol modes, for the year 2018.

Considering only the irreversible UPTK of $HNO_{3(g)}$ over dust while neglecting its UPTK over SS (the HYB_duUPTK
sensitivity run) shows very good correlation coefficients for both fine and total $NO_3^-$ concentrations (0.8 and 0.9, respectively).
Notably, the seasonal variability resembles that of experiments that completely neglect coarse $NO_3^-$ formation (fTEQ_noAlk
and fTEQ_du-ssAlk). This indicates the important role of the TEQ calculation in the fine mode, likely due to the limited
amount of sites affected significantly by nitrate coarse formation. However, a systematic positive bias results under the UPTK
assumption with values of 0.03/0.04/0.29 $\mu g\ m^{-3}$ for $HNO_{3(g)}$, fine $NO_3^-$, and Total $NO_3^-$, respectively. The correlation for
$HNO_{3(g)}$ is much lower (0.23), but still within the upper range compared to all sensitivity runs. Results from HYB_duUPTK
show consistency across continents, with notable overestimation of Total $NO_3^-$ over Asia in November and over Europe from
May to September, likely influenced by dust events affecting the monitoring sites.

By extending the UPTK reactions to include SS (sensitivity run HYB_du-ssUPTK), the previously identified overestimation
is exacerbated due to a excessive condensation of $HNO_{3(g)}$ into coarse $NO_3^-$ compared to observations. Specifically, the
correlation for Total $NO_3^-$ decreases significantly from 0.9 to 0.66. Under HYB_du-ssUPTK, $HNO_{3(g)}$ concentrations are
underestimated with a bias of -0.10 $\mu g\ m^{-3}$, while Total $NO_3^-$ surface concentrations are overestimated by 1.50 $\mu g\ m^{-3}$,
which represents the largest bias among sensitivity runs (as shown in Table 4). This positive bias in $NO_3^-$ is mainly observed
over Asia (3-5 $\mu g\ m^{-3}$ vs. 1 $\mu g\ m^{-3}$) and North-Central America (1.5-2.5 $\mu g\ m^{-3}$ vs. 1.0 $\mu g\ m^{-3}$), while differences in
European sites are more pronounced during the northern hemisphere summer and fall (Figure S12). Additionally, modifying
the value of $\gamma(HNO_3)$ for dust and keeping it constant (run HYB_g0p1) further enhances the overestimation of total $NO_3^-$ by
an additional 30%.

To evaluate whether the excessive $NO_3^-$ overestimation is due to unlimited gas availability for condensation in each size
mode, we evaluate the results obtained from the HYB_DL configuration. Surface concentrations in HYB_DL show minimal





change compared to HYB_du-ssUPTK, with the bias in total $NO_3^-$ decreasing marginally from 1.5 to 1.3 $\mu g\ m^{-3}$ (see Table S7 and Figure S14). Overall, the adoption of large values for $\gamma(HNO_3)$ for SS particles in this study may explain the overly efficient nitrate formation found in HYB_du-ssUPTK, which suggests the necessity for further refinement of HYB schemes.

The systematic overestimation of $NO_3^-$ in HYB schemes is markedly reduced when adopting a reversible partitioning of the semi-volatile inorganic species in the full range of particle sizes (DBCLL scheme). Specifically, the bias in total $NO_3^-$ surface concentrations is shifted from 0.29 to -0.43 $\mu g\ m^{-3}$ between the HYB and DBCLL runs neglecting alkalinity, and remarkably reduced from 1.5 to -0.1 $\mu g\ m^{-3}$ between HYB and DBCLL runs incorporating dust and SS NVC (Table 4), while maintaining a high correlation of around 0.8.

If dust and SS NVC are excluded in the TEQ reactions in a DBCLL mechanism (DBCLL_noAlk run), $HNO_{3(g)}$ is generally overestimated with a bias of 0.13 $\mu g\ m^{-3}$ and its seasonal cycle is poorly captured, with very low correlations of 0.06. Surprisingly, fTEQ runs show better results despite completely neglecting coarse $NO_3^-$ formation. The positive biases in $HNO_{3(g)}$ are particularly dominated by a consistent overprediction throughout the year over Asia and during specific periods over Europe and Central-North America. Excessive levels of $HNO_{3(g)}$ are accompanied with a slight underprediction in total $NO_3^-$, with a bias of -0.43$\mu g\ m^{-3}$. Including solely dust alkalinity in the DBCLL mechanism by implementing reactions R6-11 from Table 1 (DBCLL_duAlk), results in opposite results: $HNO_{3(g)}$ is underestimated with respect to observations from March to September by -0.1$\mu g\ m^{-3}$, while both fine and total particulate $NO_3^-$ are overestimated with respect to observations by 1.5 and 2 $\mu g\ m^{-3}$ respectively (see Figure S14). These biases for $HNO_{3(g)}$ and particulate $NO_3^-$ are improved when the effect of SS alkalinity is additionally considered in the formation of $NO_3^-$ (R6-12 from Table 1; DBCLL_du-ssAlk simulation). The presence of dust and SS NVC increases the condensation rates of $HNO_{3(g)}$ towards fine and coarse particulate $NO_3^-$ (Figure 5), driving the results closer to observations. Namely, the DBCLL_du-ssAlk simulation reports a lower bias in total $NO_3^-$ of -0.10 $\mu g\ m^{-3}$,while maintaining a good seasonal cycle with correlation coefficients of 0.82 and 0.78 for fine and Total $NO_3^-$, respectively. This highlights the paramount importance of including $NO_3^-$ formation on SS to accurately represent $HNO_{3(g)}$ and particulate $NO_3^-$ concentrations. The sensitivity to the treatment of dust NVCs in the model can be assessed using the DBCLL_Claq run, where the Claquin et al. (1999) dataset is adopted instead of Journet et al. (2014) (see Sect. 2.2.3). The evaluation revels limited differences compared to DBCLL_du-ssAlk, although both correlation and bias are slightly improved for total $NO_3^-$ with 0.81 and -0.09 $\mu g\ m^{-3}$, respectively (Table S7). Given the limited number of observation sites used in our evaluation, the impact of dust NVC representation appears to be significant.

### 3.2.2 Ammonia and particulate ammonium

Results for $NH_{3(g)}$ and particulate $NH_4^+$ are more consistent across mechanisms compared to $NO_3^-$ results. This consistency is expected since in all sensitivity runs use the same condensation pathway of $NH_{3(g)}$ to fine $NH_4^+$ (reactions R6 and R7 in Table 1). The seasonal cycle is reasonably well captured for both $NH_{3(g)}$ (with correlations from 0.85 to 0.88) and fine $NH_4^+$ (with correlations from 0.59 to 0.77), although the results for total $NH_4^+$ are slightly worse (correlations from 0.41 to 0.54).

The DBCLL_du-ssAlk and DBCLL_Claq runs present the lowest errors for $NH_4^+$, with RMSEs of 0.17/0.12 $\mu g\ m^{-3}$ and biases of -0.02/0.08 $\mu g\ m^{-3}$ for fine and total concentrations, respectively. The sensitivity to the treatment of dust and SS NVC





is again found to be very significant. Including or neglecting NVC representation exacerbates the biases found in the different

schemes. Unlike $NO_3^-$, the increase in particle alkalinity limits the condensation of $NH_{3(g)}$, thereby reducing the formation of

$NH_4^+$, which helps mitigate biases compared to observations.

Overall, the biases for fine and total $NH_4^+$ remain reasonably low in most experiments, ranging from -0.02 to 0.29 $\mu g\ m^{-3}$.

### 3.2.3  Sulfur species

The results for $SO_{2(g)}$ gas and particulate $SO_4^{2-}$ are consistent across all sensitivity simulations, as they are independent of

the pathways implemented for coarse $NO_3^-$ formation, with all of them forming particulate $SO_4^{2-}$ from $SO_{2(g)}$ and $H_2SO_4$

in a similar manner (see Section 2.3). Differences solely arise in the treatment of the $\gamma(SO_2)$ coefficient when considering

alkalinity of dust, which introduces differences in $SO_4^{2-}$ formation. Overall, the model slightly underestimates both $SO_{2(g)}$

and Total $SO_4^{2-}$, particularly at European stations, while it marginally overestimates fine $SO_4^{2-}$, driven by Central-North

American sites. Temporal correlations vary around 0.6, being slightly lower for Total $SO_4^{2-}$. For $SO_{2(g)}$, the small misalignment

with observations primarily stems from the European evaluation, which presents a consistent negative bias of -0.5 $\mu g\ m^{-3}$

throughout the studied period (Figure S12). Fine and total $SO_4^{2-}$ results also show acceptable agreement with observations,

with only a slight excess in fine $SO_4^{2-}$ concentrations (0.30 $\mu g\ m^{-3}$ average bias), particularly over Central and North America,

and an underestimation of total particulate $SO_4^{2-}$ surface concentrations (-0.43 average bias) over Europe (Table 4 and Figure

S12). The negative bias in $SO_4^{2-}$ may be linked to the sulfate scheme and sulfur emissions employed in our runs. Further

investigation regarding the employed $SO_{2(g)}$ uptake coefficient could be beneficial. Some alternatives to the $SO_{2(g)}$ uptake

coefficient function from Fairlie et al. (2010) have been proposed in the literature, though they lack a dependence with RH.

Possible alternatives include the uptake coefficient values proposed by Phadnis and Carmichael (2000) for $SO_{2(g)}$ uptake on

dust, and by Song and Carmichael (2001) for $SO_{2(g)}$ uptake on SS, as reported in Li et al. (2012).

### 3.2.4  Fine and total particulate matter

Regarding $PM_{2.5}$ and $PM_{10}$, results are consistent across the different sensitivity runs (Figure 5), indicating that the total mass

is dominated by other aerosol components other than secondary inorganic species. In all cases, the global $PM_{2.5}$ seasonal cycles

are well reproduced with similar correlation coefficients around 0.75. However, annual concentrations are overestimated by 20

$\mu g\ m^{-3}$ on average (Table 4). Note that the comparison is clearly driven by the dominant presence of stations over Asia, where

high concentrations are present. Conversely, results over Europe and North and Central America show a slight underestimation

of $PM_{2.5}$ concentrations.

A similar pattern is found for $PM_{10}$, though North and Central America surface concentrations are underestimated, specially

from April to September, which does not seem to be a caused by the underestimation of any secondary inorganic aerosol species

studied in this work. Consequently, $PM_{10}$ RMSE (ranging from 22.49 to 30.42 $\mu g\ m^{-3}$) and correlation coefficients (from

0.36 to 0.46) are slightly worsened compared to $PM_{2.5}$ (RMSE from 24.74 to 26.16 $\mu g\ m^{-3}$ and correlations from 0.74 to

0.76, Table 4). Consistent results have been reported by Jones et al. (2021) over Europe. We estimate that the $PM_{10}$ fraction

of secondary inorganic aerosols condensing on dust and SS accounts for up to 25% of the total $PM_{10}$ mass. This estimation



**Figure 5.** Observational evaluation of gas and particulate species. Black solid dots and crosses represent monthly median and mean of observations, respectively. Colored lines represent each configuration's monthly median surface concentrations over observational points. Error bars are the observational interquantile 0.25 to 0.75 distance. Blue shading is the interquantile 0.25 - 0.75 distance for the DBCLL_du-ssAlk simulation. For the Total $NO_3^-$ modes, data from Europe, Asia and North-Central America have been averaged, despite data from Asia and North-Central America referring to total particle concentration, while data from Europe is limited strictly to below $10 \mu m$ particle diameter.

is derived from noHC run, which excludes heterogeneous chemistry effects on nitrates and ammonia but includes secondary particles typical of other methodologies.





**Table 4:** Correlation coefficients, bias and root mean square error (RMSE) of the median of each configuration with respect to the median of observations for timeseries of Figure 5.

| | $HNO_3$ | | | $PM_{2.5}NO_3$ | | | **Total** $NO_3$ | | |
|---|---|---|---|---|---|---|---|---|---|
| | **corr** | **bias** | **rmse** | **corr** | **bias** | **rmse** | **corr** | **bias** | **rmse** |
| **noHC** | 0.03 | 0.15 | 0.20 | 0.00 | -0.43 | 0.49 | 0.00 | -0.70 | 0.77 |
| **fTEQ_noAlk** | 0.15 | 0.08 | 0.12 | 0.70 | -0.10 | 0.23 | 0.91 | -0.19 | 0.25 |
| **fTEQ_du-ssAlk** | 0.23 | 0.07 | 0.10 | 0.76 | 0.02 | 0.20 | 0.88 | -0.09 | 0.19 |
| **HYB_duUPTK** | 0.23 | 0.03 | 0.06 | 0.80 | 0.04 | 0.18 | 0.90 | 0.29 | 0.36 |
| **HYB_du-ssUPTK** | 0.06 | -0.10 | 0.11 | 0.77 | 0.14 | 0.21 | 0.66 | 1.50 | 1.56 |
| **DBCLL_noAlk** | 0.06 | 0.13 | 0.16 | 0.68 | -0.31 | 0.36 | 0.86 | -0.43 | 0.46 |
| **DBCLL_du-ssAlk** | 0.16 | 0.08 | 0.12 | 0.82 | -0.18 | 0.24 | 0.78 | -0.10 | 0.23 |

| | $NH_3$ | | | $PM_{2.5}NH_4$ | | | **Total** $NH_4$ | | |
|---|---|---|---|---|---|---|---|---|---|
| | **corr** | **bias** | **rmse** | **corr** | **bias** | **rmse** | **corr** | **bias** | **rmse** |
| **noHC** | 0.80 | -0.29 | 0.44 | 0.00 | -0.48 | 0.51 | 0.00 | -0.45 | 0.46 |
| **fTEQ_noAlk** | 0.88 | -0.65 | 0.69 | 0.64 | 0.29 | 0.35 | 0.41 | 0.26 | 0.28 |
| **fTEQ_du-ssAlk** | 0.87 | -0.62 | 0.67 | 0.59 | 0.18 | 0.28 | 0.51 | 0.17 | 0.20 |
| **HYB_duUPTK** | 0.87 | -0.61 | 0.66 | 0.56 | 0.16 | 0.27 | 0.48 | 0.15 | 0.18 |
| **HYB_du-ssUPTK** | 0.86 | -0.55 | 0.60 | 0.70 | 0.03 | 0.16 | 0.54 | 0.10 | 0.13 |
| **DBCLL_noAlk** | 0.86 | -0.58 | 0.62 | 0.66 | 0.10 | 0.20 | 0.51 | 0.16 | 0.18 |
| **DBCLL_du-ssAlk** | 0.85 | -0.53 | 0.59 | 0.70 | -0.02 | 0.17 | 0.47 | 0.08 | 0.12 |

| | $SO_2$ | | | $PM_{2.5}SO_4$ | | | **Total** $SO_4$ | | |
|---|---|---|---|---|---|---|---|---|---|
| | **corr** | **bias** | **rmse** | **corr** | **bias** | **rmse** | **corr** | **bias** | **rmse** |
| **noHC** | 0.61 | 0.65 | 1.24 | 0.59 | -0.79 | 0.80 | 0.62 | -1.91 | 1.93 |
| **fTEQ_noAlk** | 0.59 | 0.23 | 1.12 | 0.61 | 0.33 | 0.36 | 0.51 | -0.47 | 0.58 |
| **fTEQ_du-ssAlk** | 0.63 | -0.30 | 0.91 | 0.57 | 0.30 | 0.33 | 0.60 | -0.38 | 0.49 |
| **HYB_duUPTK** | 0.63 | -0.29 | 0.91 | 0.57 | 0.29 | 0.33 | 0.59 | -0.38 | 0.50 |
| **HYB_du-ssUPTK** | 0.63 | -0.29 | 0.91 | 0.57 | 0.29 | 0.33 | 0.60 | -0.38 | 0.49 |
| **DBCLL_noAlk** | 0.59 | 0.23 | 1.12 | 0.58 | 0.32 | 0.36 | 0.49 | -0.47 | 0.59 |
| **DBCLL_du-ssAlk** | 0.63 | -0.29 | 0.91 | 0.56 | 0.29 | 0.32 | 0.69 | -0.52 | 0.67 |





**Table 4 continued from previous page**

|  | PM$_{2.5}$ | | | PM$_{10}$ | | |
|---|---|---|---|---|---|---|
|  | **corr** | **bias** | **rmse** | **corr** | **bias** | **rmse** |
| **noHC** | 0.75 | -5.62 | 9.55 | 0.40 | -23.16 | 28.42 |
| **fTEQ_noAlk** | 0.76 | 20.91 | 24.98 | 0.46 | 5.14 | 22.49 |
| **fTEQ_du-ssAlk** | 0.74 | 21.75 | 26.16 | 0.44 | 7.16 | 24.38 |
| **HYB_duUPTK** | 0.74 | 21.05 | 25.32 | 0.40 | 10.16 | 27.41 |
| **HYB_du-ssUPTK** | 0.76 | 19.59 | 23.91 | 0.36 | 13.92 | 30.45 |
| **DBCLL_noAlk** | 0.76 | 20.59 | 24.74 | 0.46 | 4.98 | 22.50 |
| **DBCLL_du-ssAlk** | 0.75 | 21.02 | 25.22 | 0.43 | 8.09 | 24.80 |

## 3.3 Nitrogen partitioning

In this section, we explore the influence of coarse particulate $NO_3^-$ formation on the overall partitioning of atmospheric nitrogen

into gas and particle phases, as well as its distribution between oxidized and reduced forms.

### 3.3.1 Partitioning between gas and particle phases

Simulated nitrogen compounds in the gas and aerosol phases in the atmosphere vary depending on the underlying chemistry assumptions. We analyze the average atmospheric burden and the accumulated deposition of all nitrogen species considered in the MONARCH model across different sensitivity runs, as shown in Figure 6 and Table S13. The simulation labeled as

noHC, which excludes heterogeneous chemistry involving nitrate, serves as the baseline for understanding the distribution of nitrogen species in the gas phase. Under this assumption, the predominant nitrogen species in the atmosphere include $HNO_{3(g)}$, peroxyacetyl nitrate (PAN), $NH_{3(g)}$, and $NO_x$, with burdens of 1.06, 0.8, 0.71, and 0.38 TgN, respectively. $HNO_{3(g)}$ and PAN are characterized by longer atmospheric lifetimes compared to $NO_x$ due to their chemical stability and lower reactivity. While other species have a minor presence in the atmosphere, $N_2O_{5(g)}$ is particularly relevant because it enhances the production

of $HNO_{3(g)}$ during nighttime through heterogeneous hydrolysis (Riemer et al., 2003). Overall, the total nitrogen burden in the atmosphere is highly sensitive to the adopted nitrate mechanism, resulting in lower burdens in the schemes that are more efficient in producing $NO_3^-$ in the aerosol phase, particularly in the coarse mode.

Assuming irreversible condensation of $HNO_{3(g)}$ in coarse particles results in a reduction of the nitrogen load of 32% and 40% compared to the baseline noHC if only dust (HYB_duUPTK) or both dust and SS (HYB_du-ssUPTK) are considered,

respectively. In particular, in the HYB_duUPTK simulation, a significant portion of $HNO_{3(g)}$ partitions into the particle phase, accounting for 0.13 TgN of fine and coarse $NO_3^-$ on dust. This amount increases significantly when SS is included, with an additional consumption of $HNO_{3(g)}$. Thus, the size distribution of particulate $NO_3^-$ shifts towards the coarse mode (0.27 TgN), while fine $NO_3^-$ remains almost unaffected (0.12 TgN) compared to HYB_duAlk (Table S13).

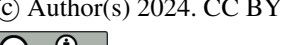

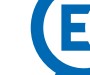

**Figure 6.** Nitrogen atmospheric 2018 average burdens (top) and accumulated depositions (bottom) of the nitrogen-containing species simulated in MONARCH for the different dust heterogeneous chemistry mechanisms. Total nitrogen deposition is reported on the top of each bar of the depositions plot.



To elucidate the impact of $HNO_{3(g)}$ UPTK by dust and SS particles, we compare the nitrogen burdens from HYB_duUPTK
and HYB_du-ssUPTK runs against those from fTEQ_du-ssAlk: results remark the role of dust and SS in decreasing $HNO_{3(g)}$
gas phase burdens by 0.21 TgN on dust and 0.50 TgN on dust and SS, compared to its sole condensation in the fine mode
through TEQ, therefore driven by the irreversible UPTK of $HNO_{3(g)}$ in the coarse mode. In this context, budget analysis
reveals that $HNO_{3(g)}$ UPTK on SS significantly alters the deposition rates of $NO_3^-$, increasing total deposition from 80 to
140 Tg/y and reducing its atmospheric lifetime from 2.6 to 2.3 days (Table 5). The combined UPTK on dust and SS nearly
depletes $HNO_{3(g)}$, leaving only 0.15 TgN remaining, which results in an overestimation of surface $NO_3^-$ as discussed in
Sections 3.2. In addition, the irreversible condensation of $HNO_{3(g)}$ on dust and SS particles also affects other gas-phase spices.
Specifically $NO_{(g)}$, $NO_{2(g)}$ and $N_2O_{5(g)}$ burdens decrease to 0.11, 0.17 and 0.01 TgN, respectively. This reduction in $NO_{x(g)}$
can be attributed to increased $NO_3^-$ particle formation, which enhances the total particle concentration and, consequently, the
available surface where hydrolysis of $N_2O_{5(g)}$ can take place. Ultimately, it produces more $HNO_{3(g)}$ available for partitioning
into the particle phase. Among the mechanisms analyzed, HYB_g0p1 shows the most efficient depletion of $HNO_{3(g)}$, achieved
by using a constant value of $\gamma = 0.1$ instead of the RH-dependent dust uptake coefficient from Fairlie et al. (2010). In this
scenario, $HNO_{3(g)}$ is further reduced to 0.46 TgN, accompanied by an increase in coarse $NO_3^-$ formation up to 1.93 Tg.
Furthermore, the deposition rates rise from 140 to 149 Tg/y, as shown in Table 5.

Conversely, nitrogen loads increase when reversible condensation of $HNO_{3(g)}$ and $NH_{3(g)}$ is considered in the fTEQ and
DBCLL sensitivity simulations. A larger proportion of nitrogen remains in the gas phase compared to HYB schemes. Notably,
$HNO_{3(g)}$ load slightly increases relative to the noHC baseline (from 1.11 to 1.14 Tg N) when dust and SS alkalinity is ne-
glected in the partitioning (fTEQ_noAlk and DBCLL_noAlk runs). Furthermore, $NO_{(g)}$ and $NO_{2(g)}$ increase by 0.29% and
0.12% respectively, and PAN decreases by nearly 25%. Conversely, accounting for the alkalinity in the reversible partitioning
(fTEQ_du-ssAlk, DBCLL_duAlk and DBCLL_du-ssAlk) generally results in higher condensation rates of $HNO_{3(g)}$ (reducing
its loads by 40% and 50% compared to noHC) to form particulate $NO_3^-$. This shift towards the aerosol phase is an expected
consequence of the neutralization of NVC from dust and SS by $HNO_{3(g)}$. Additionally, $NO_{(g)}$, $NO_{2(g)}$ and $N_2O_{5(g)}$ budgets
decrease by approximately 20% compared to analogous mechanisms without alkalinity, driven by the more efficient $N_2O_{5(g)}$
aqueous dissociation due to the increased particle presence. SS alkalinity alters the partitioning of aerosol $NO_3^-$ from fine
(-72%) towards coarse mode (+63%) when comparing DBCLL_duAlk and DBCLL_du-ssAlk, resulting in 0.06 and 0.18 TgN
of $HNO_{3(g)}$, respectively (see DBCLL_du-ssAlk run in Figure 6). This surge in coarse $NO_3^-$ formation causes a 26% decrease
of total $NO_3^-$ burdens, corresponding to a 14% increase in $NO_3^-$ deposition rates compared to the DBCLL_duAlk case, mainly
driven by enhanced wet deposition of coarse $NO_3^-$ from 11.0 to 27.5 Tg/y (Table 5). This can be attributed to the higher depo-
sition efficiency of SS particles due their larger size ranges and abundance near the surface. Consequently, the presence of SS
decreases the atmospheric lifetime of particulate $NO_3^-$ from 4.9 to 3.1 days (Table 5), contrasting with scenarios that consider
only dust alkalinity in coarse $NO_3^-$ formation. Finally, $NH_{3(g)}$ and $NH_4^+$ partitioning are also very sensitive to the inclusion
of SS NVC, leading to a partial inhibition of $NH_{3(g)}$ condensation into $NH_4^+$ (from 0.31 to 0.22 Tg, Table S11). This effect is
likely due to SS contributing additional basic NVC that hinders the formation of particulate $NH_4^+$.





### 3.3.2 Reduced and oxidized nitrogen

The AeroCom phase III nitrate experiment (Bian et al., 2017) provides a comprehensive range of budgets derived from different global aerosol models employing mechanisms similar to those analyzed in our study. Here, we compare the partitioning sensitivities in terms of oxidized and reduced nitrogen with the global estimates reported in AeroCom. In particular, the results of the EMAC model from the nitrate phase III AeroCom experiment are reported separately due to their similarity to the DB-CLL mechanism adopted in our analysis. Additionally, the study by Hauglustaine et al. (2014), which utilizes the LMDz-INCA model with an irreversible uptake mechanism, provides a reference for the HYB runs. Figure 7 presents the annual averages of oxidized and reduced nitrogen burdens and deposition for each sensitivity run.

HYB mechanisms yield a total annual average burden of oxidized species (0.55 TgN) that is slightly below the mean of AeroCom results but align well with Hauglustaine et al. (2014). Oxidized nitrogen budgets in these mechanisms fall within the lower range compared to the mechanisms that account for alkalinity (0.6 TgN vs. average 0.8 TgN in fTEQ and DBCLL). This may be attributed to an efficient deposition of nitrogen in the form of particulate $NO_3^-$ and the limited amount of $HNO_{3(g)}$ left in the atmosphere under the irreversibility assumption already noted in previous Sections 3.2 and 3.3.1. Regarding reduced nitrogen, the budgets from the HYB mechanisms (approx. 0.3 TgN) align well with both the AeroCom mean and Hauglustaine et al. (2014), although they present a relatively lower fraction of particulate $NH_4^+$. The $NH_{3(g)}$ budgets show an inverse relationship with the concentration of $HNO_{3(g)}$, indicating that $HNO_{3(g)}$ acts as a limiting factor for the $NH_{3(g)}$ neutralization in the fine mode. This relationship corresponds with the slightly lower $NH_4^+$ budgets observed in HYB simulations (approx. 0.15 TgN, see also Table S11). Overall, this sheds light on the dynamics of $NH_{3(g)}$ with sulfate, suggesting that the mass of $NH_4^+$ formed in the HYB mechanisms is mainly a product of $NH_{3(g)}$ neutralization with sulfate through the fTEQ calculation, given that most $HNO_{3(g)}$ is consumed by the UPTK reactions.

The fTEQ and DBCLL mechanisms exhibit a high sensitivity in the partitioning of oxidized and reduced nitrogen to alkalinity, regardless of whether coarse $NO_3^-$ formation is considered. When dust and SS alkalinity are neglected (fTEQ_noAlk and DBCLL_noAlk runs), the burdens of oxidized nitrogen are notably high (1.1 TgN). This primarily consists of unreacted $HNO_{3(g)}$, which even slightly exceeds those obtained from the run without heterogeneous chemistry (noHC, 1.05 TgN). This result is attributed to the sensitivity of $N_2O_{5(g)}$ hydrolisis under different aerosol loading conditions. Conversely, the burdens of reduced nitrogen in these schemes remain relatively low compared to other mechanisms. This is mainly due to the high consumption of $NH_{3(g)}$ by neutralization of $HNO_{3(g)}$, which acts as an important sink of reduced nitrogen as particulate $NH_4^+$. The introduction of dust and SS NVC in the reactions decreases oxidized nitrogen by an enhanced consumption of $HNO_{3(g)}$ to form $NO_3^-$. Furthermore, reduced nitrogen concentrations increase as a consequence of the lower availability of $HNO_{3(g)}$ to neutralize $NH_{3(g)}$. This phenomenon is similarly observed in simulations including TEQ calculations over fine (fTEQ_dussAlk) and over both fine and coarse modes (DBCLL_duAlk and DBCLL_du-ssAlk). Comparatively, the presence of dust and SS alkalinity decreases the oxidized nitrogen burdens by approximately 27% compared to analogous simulations neglecting NVC. However, contrasting the results between DBCLL_duAlk and DBCLL_du-ssAlk shows that while the presence of SS NVC further enhances the consumption of $HNO_{3(g)}$ by 20%, it concurrently reduces $NO_3^-$ burdens by 5%. This is likely due





to SS contributing with acidic NVC such as $Cl^-$ and $SO_4^{2-}$ (see Section 2.2.3), which hinders the formation of particulate $NO_3^-$.

Alkalinity leads to higher reduced nitrogen budgets due to enhanced basic conditions (high pH, as shown in Figure 4) facilitated by dust and SS NVC. This environment inhibits $NH_{3(g)}$ condensation over dust and SS particles, thereby increasing the $NH_{3(g)}$ atmospheric lifetime (Table S10) and allowing it to mix with sulfate from polluted areas, as can be seen in the spatial correlation between particulate $SO_4^{2-}$ and $NH_4^+$ (see discussion in Section 3.1 and Figures S4 and S7). Consequently, alkalinity facilitates the formation of $NH_4^+$ through the neutralization of $NH_{3(g)}$ with sulfate, contrasting with the neutralization process

involving $HNO_{3(g)}$ observed in the DBCLL_noAlk simulation. Overall, the DBCLL_du-ssAlk run aligns with the average of AeroCom in both oxidized and reduced partitions, although its reduced phase falls below EMAC levels. This discrepancy can be attributed to EMAC showing very low deposition rates of $NH_{3(g)}$ (see Table S10), which enhances its atmospheric burden.

### 3.4    Nitrate budgets

The $NO_3^-$ budgets from the sensitivity runs are summarized in Table 5, including burdens, wet and dry deposition, produc-

tivities, and lifetime of fine, coarse and total $NO_3^-$. Comparative data from previous studies (Bian et al., 2017; Karydis et al., 2016; Rémy et al., 2022; Hauglustaine et al., 2014; Jones et al., 2021) are also included. Analogous information on $HNO_{3(g)}$, $NH_{3(g)}$, total $NH_4^+$, and $SO_4^{2-}$ can be found in the Supplementary Section S4 (Tables S9 to S12).

The sensitivity runs conducted in this study yield a wide range of total $NO_3^-$ burdens, ranging from 0.09 to 1.93 Tg. Experiments neglecting coarse $NO_3^-$ and the role of non-volatile components (NVC) fall on the lower end of this spectrum, while

the HYB schemes consistently report higher burdens. The fine to coarse ratio increases in those mechanisms that neglect part or completely NVC in the nitrate partitioning, and HYB runs consistently simulate the highest coarse burdens (0.57 to 1.43 Tg). Compared with the reported values in the literature, both HYB (1.15 to 1.93 Tg) and DBCLL (1.04 to 1.44 Tg) schemes considering NVC fall within the uper range of Aerocom total $NO_3^-$ burden intercomparison study (Bian et al., 2017) and above specific global systems such as IFS (Rémy et al., 2022) or MetOffice UM (Jones et al., 2021). Similar differences are found in

total annual depositions with ranges that span a factor of two among schemes. For wet deposition, DBCLL mechanisms show values ranging from 37 to 45 Tg/y, within the lower half of the range found in the literature (45±30 Tg/y for AeroCom to 75 Tg/y for IFS), while HYB consistently simulate higher estimates even beyond reported values (63 to 120 Tg). This feature is also seen in the dry deposition. Interestingly, the lifetimes of total $NO_3^-$ shown in Table 5 range from 2.3 to 2.6 days for global models adopting HYB schemes, systematically estimating the lowest lifetimes, while models introducing reversible partition-

ing provide longer lifetimes (2.6 to 4.9 days). To better understand the possible reasons for such a wide range of estimates, we analyze some of our sensitivity runs compared to similar systems in the literature.

The assumption of irreversibility of nitrate condensation onto dust and SS particles (HYB_du-ssUPTK run) produces the largest increase in coarse $NO_3^-$ formation, exceeding values reported in the literature. For instance, the HYB_du-ssUPTK model reports a total $NO_3^-$ burden of 1.75 Tg, which is higher than the AeroCom range (0.63 ± 0.56 Tg) and values from

models such as EMAC in AeroCom (Bian et al., 2017), as well as findings from Bian et al. (2017), Jones et al. (2021), Rémy et al. (2022) and Hauglustaine et al. (2014) (0.67, 0.89, 0.82, and 0.80 Tg, respectively). Additionally, the results surpass those




**Figure 7.** Average 2018 budgets (top) and depositions (bottom) for the different dust heterogeneous chemistry mechanisms. In each plot, reduced and oxidized species for each mechanism (columns) are compared to the references (three right columns).





**Table 5.** Results for fine, coarse and total particulate $NO_3^-$ obtained with the studied heterogeneous chemistry mechanisms. Results from the references are reported at the end of the Table: the average of all the participating models in the intercomparison AeroCom phase III nitrate experiment for 2008, specifying their standard deviation (*STD*), and results from the GMI model (using a similar HYB approach with UPTK reactions on dust and SS) and the EMAC model (*EMAC 2008*, that uses a similar approach to DBCLL_du-ssAlk). Also using the EMAC model, results obtained by the Karydis et al. (2016) study are reported as *EMAC 2005-2008*. Results from models using a similar approach to HYB_du-ssUPTK are reported as *IFS* for Rémy et al. (2022), *LMDz-INCA* for Hauglustaine et al. (2014) and *MetOffice UM* for Jones et al. (2021).

| $NO_3^-$ | Burden (Tg) | | | Wet Dep. (Tg y-1) | | | Dry Dep. (Tg y-1) | | | Total Dep. (Tg y-1) | | | Production (Tg y-1) | | | Lifetime (days) | | |
|---|---|---|---|---|---|---|---|---|---|---|---|---|---|---|---|---|---|---|
| **Experiment** | Fine | Coarse | **Total** | Fine | Coarse | **Total** | Fine | Coarse | **Total** | Fine | Coarse | **Total** | Fine | Coarse | **Total** | Fine | Coarse | **Total** |
| **fTEQ_noAlk** | 0.09 | - | **0.09** | 4.6 | - | **4.6** | 2.0 | - | **2.0** | 6.6 | - | **6.6** | 6.6 | - | **6.6** | 2.6 | 0.0 | **2.6** |
| **fTEQ_du-ssAlk** | 0.66 | - | **0.66** | 36.1 | - | **36.1** | 7.4 | - | **7.4** | 43.5 | - | **43.5** | 43.6 | - | **43.6** | 2.7 | 0.0 | **2.7** |
| **HYB_duUPTK** | 0.58 | 0.57 | **1.15** | 25.8 | 37.9 | **63.7** | 5.4 | 11.1 | **16.5** | 31.2 | 49.0 | **80.2** | 31.3 | 48.9 | **80.2** | 3.4 | 2.1 | **2.6** |
| **HYB_du-ssUPTK** | 0.55 | 1.20 | **1.75** | 10.7 | 103.4 | **114.1** | 2.9 | 23.6 | **26.5** | 13.6 | 127.0 | **140.6** | 13.6 | 126.9 | **140.5** | 7.4 | 1.7 | **2.3** |
| **HYB_gdust=0.1** | 0.50 | 1.43 | **1.93** | 8.4 | 111.9 | **120.4** | 2.3 | 27.0 | **29.3** | 10.7 | 138.9 | **149.7** | 10.8 | 138.9 | **149.6** | 8.5 | 1.9 | **2.4** |
| **HYB_DL** | 0.41 | 1.26 | **1.68** | 2.8 | 109.7 | **112.6** | 1.4 | 25.7 | **27.1** | 4.2 | 135.4 | **139.6** | 4.2 | 135.3 | **139.5** | 17.8 | 1.7 | **2.2** |
| **DBCLL_noAlk** | 0.06 | 0.00 | **0.06** | 2.5 | 0.0 | **2.6** | 1.6 | 0.1 | **1.7** | 4.1 | 0.1 | **4.2** | 4.2 | 0.1 | **4.3** | 2.6 | 2.8 | **2.6** |
| **DBCLL_duAlk** | 0.96 | 0.47 | **1.44** | 26.5 | 11.0 | **37.5** | 8.3 | 8.2 | **16.5** | 34.8 | 19.3 | **54.0** | 34.7 | 19.2 | **53.9** | 5.1 | 4.5 | **4.9** |
| **DBCLL_du-ssAlk** | 0.28 | 0.78 | **1.07** | 18.2 | 27.5 | **45.7** | 4.0 | 12.6 | **16.5** | 22.2 | 40.1 | **62.3** | 22.3 | 40.0 | **62.2** | 2.3 | 3.6 | **3.1** |
| **DBCLL_ClaqAlk** | 0.41 | 0.63 | **1.04** | 17.4 | 27.8 | **45.2** | 3.8 | 12.9 | **16.7** | 21.2 | 40.7 | **61.9** | 21.3 | 40.6 | **61.9** | 3.5 | 2.8 | **3.1** |
| **AeroCom** | - | - | **0.63** | - | - | **45.9** | - | - | **20.7** | - | - | **66.6** | - | - | **60.6** | - | - | **5.0** |
| **STD AeroCom** | - | - | **±0.56** | - | - | **±30.7** | - | - | **±19.5** | - | - | **±50.2** | - | - | **±45.9** | - | - | **±2.3** |
| **GMI** | - | - | **0.97** | - | - | **43.3** | - | - | **14.8** | - | - | **58.1** | - | - | **59.3** | - | - | **6.0** |
| **EMAC 2008**[1] | - | - | **0.67** | - | - | **0.0** | - | - | **46.3** | - | - | **46.3** | - | - | **-** | - | - | **-** |
| **EMAC 2005-2008**[2] | - | - | **0.44** | - | - | **-** | - | - | **-** | - | - | **-** | - | - | **-** | - | - | **-** |
| **IFS**[3] | 0.47 | 0.35 | **0.82** | 35.9 | 39.6 | **75.5** | 4.6 | 23.8 | **28.4** | 40.5 | 63.4 | **103.9** | 40.5 | 63.4 | **103.6** | 4.2 | 2.1 | **3.0** |
| **LMDz-INCA** | 0.22 | 0.58 | **0.80** | - | - | **56.1** | - | - | **7.4** | - | - | **63.5** | 14.1 | 49.5 | **63.6** | - | - | **4.6** |
| **MetOffice UM**[4] | 0.49 | 0.40 | **0.89** | - | - | **63.3** | - | - | **39.4** | - | - | **102.8** | 27.9 | 73.5 | **101.4** | 6.2 | 2.0 | **3.2** |

1. From Bian et al. (2017).

2. From Karydis et al. (2016).

3. Fine $NO_3^-$ reported from neutralization of nitric acid, ammonia and sulfate. Coarse $NO_3^-$ from heterogeneous chemistry. (Rémy et al., 2022).

4. Jones et al. (2021) performs two sensitivity tests to the accommodation coefficient used for $NO_3^-$ formation in the fine mode: *FAST* with 0.193 and *SLOW* with 0.001. Here, results from the *FAST* test are reported on the basis that they present similar fine $NO_3^-$ formation rates to our average fine $NO_3^-$ results.





from three AeroCom models (EMEP, INCA, and GMI) using a similar HYB approach with UPTK reactions on dust and SS, which range from 0.26 to 0.95 Tg of total $NO_3^-$ (see (Bian et al., 2017)). Multiple reasons could explain some of the disagreements found in this comparison. For instance, Hauglustaine et al. (2014) employs a similar HYB implementation and UPTK
coefficients for dust as the HYB_du-ssUPTK run, although a different parametrization for the $HNO_{3(g)}$ UPTK on SS. Despite Hauglustaine et al. (2014) also overestimating particulate $NO_3^-$ with a global normalized mean bias of +68%, the HYB_du-ssUPTK simulation still reports higher burdens for $NO_3^-$ (1.75 Tg vs 0.80 Tg) and lower burdens for $HNO_{3(g)}$ (0.69 vs 1.35 Tg, Table S9) compared to Hauglustaine et al. (2014). The lower $NO_x$ emissions in our study imply a reduced availability of precursor gases, which would typically result in lower $NO_3^-$ formation (40.8 vs. 46.TgN, Table S1). However, the high uptake
coefficients employed for $HNO_{3(g)}$ UPTK on SS may be excessively efficient as discussed previously in Section 3.1.2 and 3.2. The run neglecting the uptake on SS (HYB_duUPTK) reports values closer to observations (Section 3.2) and reported budgets in the literature. Consequently, these results can be explained by the much lower UPTK coefficient for SS employed in the other systems (Hauglustaine et al., 2014; Jones et al., 2021; Rémy et al., 2022) compared to our implementation. Therefore, a function for the uptake coefficient on SS dependent on relative humidity and on SS particle size should be considered in future
works, following the same approach as for dust (equation 2) and Hauglustaine et al. (2014). However, such a function has not clearly been determined in the literature. While Hauglustaine et al. (2014) assumes a BET isotherm for the $HNO_{3(g)}$ UPTK on SS similar to the UPTK on dust (Fairlie et al., 2010), it remains unclear if this approach is suitable for SS as it is for dust. To our best knowledge, the most appropriate alternative to the assumption of averaged UPTK coefficients for SS would be to fit a function to the experimental values reported by Liu et al. (2007), which provides uptake coefficient dependencies with
relative humidity and particle size. However, several discrepancies between Liu et al. (2007) and previous studies still remain unresolved (Tolocka et al., 2004; Saul et al., 2006), presenting difficulties in reaching a common agreement on an $HNO_{3(g)}$ UPTK coefficient function on SS.

Beyond the excessive $NO_3^-$ formation driven by the UPTK on SS, the sole implementation of $HNO_{3(g)}$ UPTK on dust (HYB_duUPTK run) still slightly exceeds particulate $NO_3^-$ burdens and deposition rates reported by the references (Table
5). This is also observed when compared to observational surface concentrations (see Section 3.2), suggesting that the sole UPTK on dust results in excessive $NO_3^-$ formation. . Given that previous studies have consistently shown non negligible $HNO_{3(g)}$ uptake on SS (Myhre et al., 2006; Athanasopoulou et al., 2008), these biases could be explained by 1) an excessive UPTK coefficient employed for $HNO_{3(g)}$ UPTK on dust, 2) the inherent inappropriateness of the irreversible condensation assumption of gas species on particles to simulate particulate $NO_3^-$ formation, or 3) the different atmospheric lifetime of coarse
particles among different systems. Regarding issues in the UPTK parameterization on dust adopted in our study, the inclusion of the scaling factor for alkalinity added to the dust uptake coefficients ($Sc$ in equation 2) could contribute to excessively high UPTK rates. This alkalinity scaling factor is based on the NVC fractions provided by Fairlie et al. (2010), namely calcium and magnesium (see Section 2.2.1). Since our study includes additional NVC (i.e. potassium and sodium), this estimation might not be representative and could result in excessive condensation. However, additional information on the NVC content in Fairlie
et al. (2010) is missing to refine our approach. Concerning the irreversible condensation assumption, models implementing such an approach with similar UPTK coefficients tend to overestimate particulate $NO_3^-$ formation (Hauglustaine et al., 2014;





Jones et al., 2021; Rémy et al., 2022). Lastly, a longer lifetime of coarse particles such as dust or SS could contribute to enhance $NO_3^-$ production regardless of the adopted mechanism. Spada et al. (2013) intercompared different SS emission schemes in the MONARCH model and reported lifetimes ranging from 4 to 12 days including results from literature. These findings highlight

significant differences in emission, transport, and sedimentation schemes among global models, which are not negligible. Such differences could explain part of the variations identified in our analysis. For instance, if a model assumes a longer lifetime for coarse particles, it would result in a prolonged period during which $HNO_{3(g)}$ can condense onto these particles, thus increasing the overall burden of particulate $NO_3^-$. Understanding and standardizing these lifetimes could be crucial for achieving more consistent and accurate predictions of $NO_3^-$ aerosol formation across different modeling frameworks.

Linking with the limitations of irreversible approaches, we finally analyze the DBCLL runs to illustrate the paramount role of alkalinity in $NO_3^-$ formation. Neglecting NVC in the partitioning (DBCLL_noAlk) results in negligible $NO_3^-$ formation, with a burden of just 0.06 Tg in the fine mode. In contrast, only considering dust alkalinity (DBCLL_duAlk) significantly increases the burden of both fine and coarse modes to 0.96 and 0.47 Tg, respectively, which exceeds estimates reported in other works.

Comparing the obtained $NO_3^-$ burdens and size distributions with those reported by Rémy et al. (2022), Hauglustaine et al. (2014), and Jones et al. (2021) (Table 5), we observe that the total $NO_3^-$ values are notably high when only dust alkalinity is considered. This discrepancy can be attributed to excessive fine $NO_3^-$ formation, as already noted in the observational evaluation (see Section 3.2). This indicates that while the DBCLL approach improves the representation of alkalinity effects compared to the irreversible mechanisms, it still tends to overestimate the fine mode $NO_3^-$, suggesting a need for further

refinement in the parameterization of NVC effects to achieve more accurate nitrate aerosol predictions.

Additionally, accounting for SS alkalinity (reaction R12 from Table 1), the DBCLL_du-ssAlk run shifts particulate $NO_3^-$ formation towards the coarse mode, increasing burdens from 0.47 to 0.78 Tg. Consequently, this leads to higher total $NO_3^-$ deposition rates, rising from 54 to 62 Tg/y. Including both dust and SS alkalinity results in a notable agreement for $NO_3^-$ (1.07 Tg) and $HNO_{3(g)}$ (2.53 Tg) burdens with values reported in the literature. This improvement highlights the importance of

considering both dust and SS alkalinity in accurately modeling $NO_3^-$ formation and deposition rates in atmospheric chemistry simulations. Some differences emerge when comparing the $NO_3^-$ size distribution against references that report fine and coarse $NO_3^-$ budgets. Specifically, the fine $NO_3^-$ burden tends to be on the lower end of the references range (0.28 Tg vs 0.39 Tg on average), while the coarse $NO_3^-$ burden is on the higher end (0.78 Tg vs 0.54 Tg on average), although it should be considered the relatively high variability between the references in both modes. Despite these variations, accounting for

alkalinity significantly improves the agreement of $NO_3^-$ lifetime (3.1 days) with AeroCom (5.0±2.3 days), Rémy et al. (2022) (3.0 days) and Jones et al. (2021) (3.2 days).

To delve into the differences observed between simulations employing dust alkalinity derived from Journet et al. (2014) (DBCLL_du-ssAlk) and from Claquin et al. (1999) (DBCLL_ClaqAlk) as shown in Table 5, it becomes evident that DB-CLL_ClaqAlk results in a noticeable shift towards fine $NO_3^-$ formation (from 0.28 Tg to 0.40 Tg), with fine $NO_3^-$ increasing

its lifetime (2.3 to 3.5 days) and coarse $NO_3^-$ decreasing it (3.6 to 2.8 days). This can be attributed to the higher fraction of $Ca^{2+}$ NVC in the fine mode derived from Journet et al. (2014) (5.73%) compared to Claquin et al. (1999) (3.68%), while both



datasets present closer fractions of calcium for the coarse mode (4.61% vs. 3.74%) (see Supplementary Tables S4 and S6). This underscores the important role that $Ca^{2+}$ NVC plays in heterogeneous chemistry, surpassing the higher fractions of other NVC in the Claquin et al. (1999) dataset (0.07% Na, 0.78% K and 0.43% Mg, Tables S3 to S6). Also, we observe a slightly higher burden of $HNO_{3(g)}$ (from 2.53 Tg to 2.58 Tg) and a lower burden of $NH_{3(g)}$ (from 0.35 Tg to 0.31 Tg) (see Supplementary Tables S9 and S10). This provides a quantification of the impact that mineral distributions among dust sources (especially calcite) has on gas-aerosol partitioning, specially in the formation of particulate $NO_3^-$, and the importance of advancing on the representation of mineralogical dust composition at the global scale and its regional variability.

Finally, we compare the outcomes from the DBCLL_du-ssAlk simulation with those reported by the EMAC model, which employs a similar approach to the DBCLL mechanism and also accounts for dust and SS alkalinity (see Section 3.1.3). The analysis is performed against the two studies where EMAC budgets were reported: the AeroCom intercomparison nitrate experiment (Bian et al., 2017) and Karydis et al. (2016) (Table 5). It's notable that the total $NO_3^-$ burden (1.07 Tg) in DBCLL_du-ssAlk exceeds both the EMAC results from AeroCom (0.67 Tg) and Karydis et al. (2016) (0.44 Tg), although our findings show reasonable agreement with observations, especially over Asia (see Section 3.2). Regarding the total budget of $HNO_{3(g)}$ (2.53 Tg), despite our result closely matching the AeroCom experiment average (2.50±1.83 Tg, Table S9), it is below the values reported by EMAC in this experiment (3.10 Tg) while significantly exceeding results from Karydis et al. (2016) (1.65 Tg). These outcomes reveal intrinsic differences in the rates of $HNO_{3(g)}$ condensation on dust and SS between both models. Conversely, the $NH_{3(g)}$ burden (0.35 Tg) in our simulation falls below those reported by both studies (0.85 and 0.82 Tg, respectively), potentially due to the high formation of particulate $NO_3^-$ over dust and SS NVC in our study. Nevertheless, the resulting burdens for $NH_4^+$ align with both references (0.22 vs. 0.19 and 0.17 Tg for EMAC in AeroCom and Karydis et al. (2016), respectively). Various factors could explain the differences observed between EMAC and the DBCLL_du-ssAlk simulation: the different simulated periods between our study, Karydis et al. (2016) and AeroCom (2018 vs. 2005-2008 average and 2008, respectively), the varying NVC content assumed for dust and SS between DBCLL_du-ssAlk and EMAC (Section 3.1.3), and the intrinsic differences in the heterogeneous chemistry mechanisms employed in both models (DIFFLIM and DBCLL in MONARCH and bulk TEQ followed by DIFFLIM used in Karydis et al. (2016)). Despite their conceptual equivalence, our results demonstrate that these mechanisms may yield different outcomes.

## 4 Conclusions

In this study, we conducted a comprehensive exploration of the processes driving nitrate formation on fine and coarse particles globally using the MONARCH atmospheric model. Our sensitivity simulations incorporated state-of-the-art dust heterogeneous chemistry mechanisms, including thermodynamic equilibrium (TEQ) and irreversible uptake reactions (UPTK) between gas and aerosol phases. Three mechanisms for particulate nitrate formation were implemented: (1) fTEQ, which considers reversible condensation-evaporation focusing on fine nitrate without accounting for coarse nitrate; (2) HYB, where fine nitrate





forms via TEQ with a subsequent irreversible uptake of $HNO_{3(g)}$ on coarse particles; and (3) DBCLL, which allows for both
fine and coarse nitrate formation through a reversible process with kinetic gas limitation.

Key assumptions such as uptake coefficients, reversible partitioning, and the inclusion of dust and sea-salt alkalinity were
thoroughly assessed. Global average dust alkalinity was sourced from previous dust mineralogy simulations (Gonçalves Ageitos
et al., 2023) using Journet et al. (2014) and Claquin et al. (1999) mineral atlases, while globally homogeneous sea-salt alkalin-
ity values are derived from Seinfeld and Pandis (2006). We evaluated annual cycle surface concentrations against observations
for various atmospheric species and compared global spatial distributions and budgets with existing literature. Additionally,
we investigated the partitioning of nitrogen species and assessed the impact of dust and sea-salt heterogeneous chemistry on
nitrogen budgets.

Neglecting coarse nitrate formation (fTEQ_du-ssAlk run) results in fine nitrate budgets and distributions that closely match
observational surface concentrations of total nitrate (-0.09 $\mu g$ $m^{-3}$ bias, 0.88 correlation). This good agreement suggests
that observational stations are dominated by fine nitrate concentrations or that fine nitrate formation is overestimated in this
run. Since this mechanism aligns with references forming both fine and coarse nitrate (Bian et al., 2017; Rémy et al., 2019;
Hauglustaine et al., 2014), it likely compensates for the lack of coarse nitrate through the instantaneous TEQ of gas species in
the fine mode, thus not accurately reflecting the size distribution of particles.

The formation of coarse nitrate through the irreversible uptake of $HNO_{3(g)}$ on coarse particles (HYB methodologies) is
highly sensitive to whether the $HNO_{3(g)}$ uptake is assumed to occur solely on dust or on both dust and sea-salt particles. The
uptake solely on dust closely reproduces fine and total particulate nitrate seasonality (correlations of 0.8 and 0.9, respectively),
although it slightly overestimates observations (+0.29 $\mu g$ $m^{-3}$ bias), particularly over Asia, and nitrate budgets compared to
the literature (1.15 vs. 0.74 Tg on average). An excessively efficient $HNO_{3(g)}$ uptake on dust particles could explain such
a performance that could be improved through more accurate scaling factors adopted for alkalinity. However, the lack of
information in the literature poses a challenge in implementing such adjustments.

The introduction of the $HNO_{3(g)}$ uptake on sea-salt further increases the formation of coarse nitrate, exceeding the reported
burdens in the literature (1.75 vs. 0.74 Tg from references' average) and overestimating observational surface concentrations
(+1.50 $\mu g$ $m^{-3}$ bias). Moreover, some deviations with respect to references are observed in the latitude and magnitude of the
transatlantic transport of coarse nitrate formed during long-range transport of dust and sea-salt. These findings indicate the
need for a revision of the $HNO_{3(g)}$ uptake coefficients on sea-salt. A potential alternative could involve its implementation as a
function of relative humidity and particle size, aligning it with experimental data from Liu et al. (2007), although discrepancies
with earlier experimental studies present difficulties in determining consistent uptake coefficients for sea-salt.

The reversible condensation-evaporation of gas species on both fine and coarse modes of dust and sea-salt (DBCLL mech-
anism) highlights the paramount importance of accounting for alkalinity to derive consistent results both globally and across
continents. Remarkably, DBCLL_du-ssAlk effectively captures monthly global concentrations of fine and total nitrate, with
correlations of 0.82 and 0.78 and concentrations slightly underestimated with -0.18 and -0.10 $\mu g$ $m^{-3}$ bias, respectively. These
good results extend to particulate ammonium for both fine and total fractions (-0.02 and 0.08 $\mu g$ $m^{-3}$ bias, respectively).
Accounting for dust and sea-salt alkalinity in the DBCLL mechanism substantially increases particulate nitrate formation by




94% in comparison to the same mechanism with no alkalinity, and decreases the reduced nitrogen burden due to an increase of ammonium formation. Moreover, the sole inclusion of sea-salt alkalinity is identified as responsible of substantially shifting the size partitioning of particulate nitrate from fine (-72%) to coarse (+63%), along with the rise in wet deposition of particulate $NO_{3(a)}$ due to high scavenging rate of sea-salt particles, overall reducing the total aerosol nitrate atmospheric lifetime from 4.9 to 3.1 days.

A comparison between runs adopting the two different dust alkalinity fractions derived from averaging simulations using Claquin et al. (1999) and Journet et al. (2014) soil mineralogy datasets as described in Gonçalves Ageitos et al. (2023) reveals that assumptions on the dust composition are crucial. A decrease from 5.17% to 3.68% in Ca along with slight increases in Na, K and Mg between Journet et al. (2014) and Claquin et al. (1999) leads to a +35% increase in fine and -21% decrease in coarse nitrate formation. This high sensitivity is especially attributed to the lower calcite content in the Claquin et al. (1999) database.

Comparison with references reveals that while surface concentrations in the DBCLL_du-ssAlk simulation align well with observations, its global burden (1.07 Tg) sits at the upper limit of AeroCom's reported range (0.63±0.56 Tg) and slightly exceeds the average reported by other studies (0.74 Tg). Fundamental differences between the models, like the alkalinity factors, intrinsic differences in the heterogeneous chemistry, aerosol representation and transport processes can partly explain the wide range of results reported in the literature.

This study establishes a crucial benchmark for future investigations into the impact of incorporating regional variations in dust alkalinity, as derived from Claquin et al. (1999), Journet et al. (2014), and the upcoming spectroscopically-based EMIT surface mineralogical dataset (Green et al., 2018; Thompson et al., 2024; Brodrick et al., 2023), on the formation of particulate nitrate and atmospheric composition. The findings also evaluate the significance of dust and sea-salt alkalinity in inorganic aerosol heterogeneous chemistry, providing insights into the optimal representation of dust alkalinity in atmospheric models. Ultimately, these results aim to enhance the capability of atmospheric and climate models to simulate the formation of aerosol nitrate, ammonium, and sulfate, potentially improving our ability to estimate the radiative effects of these species in climate projections.

*Code availability.* The MONARCH code is available at https://earth.bsc.es/gitlab/es/monarch (last access: 21 July 2024) and the the HERMESv3_GR code is accessible at https://earth.bsc.es/gitlab/es/hermesv3_gr (last access: 21 July 2024).

*Data availability.* The GHOST dataset is made freely available via the following repository: https://doi.org/10.5281/zenodo.10637449 (Bowdalo et al., 2024). The model output used in this work is available in the Zenodo data repository at https://doi.org/10.5281/zenodo.12789730 (Sousse, 2024).



*Author contributions.* RS, OJ and CPG-P developed the model, designed the methodology and the conceptualization, and performed the investigation and analysis of the results. RS designed and conducted the sensitivity simulations, and the results' postprocessing, validation and visualization, assisted by OJ and CP. MGA provided the simulations with the average mineralogy from the utilized mineral dust datasets.

DB provided the observational evaluation data under request, developed the GHOST dataset and the evaluation software. MGV provided the CAMS-ANTv4.2 emission dataset under request and developed the HERMESv3_GR emission model. RS wrote the manuscript, which was re-edited by OJ and CPG-P, with contributions from all other co-authors.

*Competing interests.* The authors declare that they have no competing interests.

*Acknowledgements.* This work was funded by the European Research Council (ERC) under the Horizon 2020 research and innovation pro-
1085 gram through the ERC Consolidator Grant FRAGMENT (grant agreement no. 773051). We additionally acknowledge support from the AXA Research Fund through the AXA Chair on Sand and Dust Storms at the Barcelona Supercomputing Center (BSC), the Spanish Ministerio de Economía y Competitividad through the HEAVY project (grant PID2022-140365OB-I funded by MICIU/AEI/10.13039/501100011033 and by ERDF, EU), the European Union's Horizon 2020 research and innovation program under grant agreement no. 821205 (FORCeS) and the Department of Research and Universities of the Government of Catalonia via the Research Group Atmospheric Composition (code 2021
SGR 01550). RS was funded by the predoctoral program AGAUR-FI ajuts (2023 FI-3 00065) Joan Oró, which is backed by the Secretariat of Universities and Research of the Department of Research and Universities of the Generalitat of Catalonia, as well as the European Social Plus Fund.

The authors acknowledge the computer resources at Marenostrum and the technical support provided by Barcelona Supercomputing Center (RES-AECT-2022-3-0013, RES-AECT-2023-2-0008, RES-AECT-2023-3-0026), with special mention to Alejandro García, Carles
Tena, Gilbert Montane and Albert Vila.



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
