# Peer review of "A Comprehensive Global Modelling Assessment of Nitrate Heterogeneous Formation on Desert Dust"

_EGUsphere, 2024_

## Author Response (AR2)

**REFEREE #1**

Review of Rubén Soussé Villa et al.: "A Comprehensive Global Modelling Assessment of Nitrate Heterogeneous Formation on Desert Dust"

*This paper investigates the processes driving nitrate formation on fine and coarse particles on a global scale, using the MONARCH global atmospheric chemistry transport model. It specifically focuses on the key processes involved in nitrate formation over dust and evaluates their representation within the model. The study integrates varying levels of complexity in dust heterogeneous chemistry into the MONARCH model. Three main mechanisms for particulate nitrate formation were implemented: fTEQ, HYB, and DBCLL. The methodologies incorporate various assumptions, including uptake coefficients, reversible partitioning, and the influence of dust and sea-salt alkalinity. The study further indicates that the formation of coarse nitrate through the irreversible uptake of $HNO_{3(g)}$ on coarse particles is highly sensitive to whether it occurs solely on dust or on both dust and sea-salt particles. The analysis emphasizes the implications of nitrate formation on burdens and the role of alkalinity. The findings show differences based on the selected methodology, with a broad range of burdens for the particulate nitrate and the correlations with observations. Overall, the authors highlight the importance of incorporating dust and sea-salt alkalinity into global nitrate simulations along with thermodynamic processes, which were found to be more aligned with observational data.*

**General comments:**

*The manuscript provides a clear description of its objectives and is well written. I acknowledge the authors' effort in presenting the numerous sensitivity simulations; however, the lengthy discussion on the differences in these sensitivity simulations used to incorporate nitrate particles into the model may be challenging for the reader (e.g., many abbreviations concerning the various subcases, etc.) to grasp the significance of the results. Some simulations' analysis could be included in the supplementary material and only briefly discussed in the main text to better emphasize the primary findings of this study. Aside from this minor concern, the discussion of the results is generally very well organized, though some repetition is evident at the beginning of some sections. I don't have any major comments, but there are a few minor issues regarding the modeling method that can be discussed to support the results and conclusions of this study. Therefore, I recommend a revision to address these issues before the acceptance of the submission.*

We thank the reviewer for the general positive feedback. Following the reviewer's suggestions, we moved the discussion of simulations HYB_g0p1, DBCLL_duAlk and DBCLL_ClaqAlk to the Supplementary Section S5. This affects Sections 3.1.2 and 3.2.1, reducing their length and easing their readability.

The main text of Section 3 was also modified to enhance clarity and readability, summarize some parts, removing the description of some experiments to avoid repetition, adding more references, and highlighting the main conclusions from each section.

We believe that the revised version, without changing the original message, improves the readability of the manuscript.

**Minor comments:**

*Line 335: It is not clear why the 50% fraction of $H_2SO_{4(g)}$ is applied. Can the authors provide some evidence for this fraction?*

The MONARCH atmospheric chemistry model adopts a mass-based aerosol representation designed to reproduce the aerosol concentrations and burdens in the atmosphere. The model does not include detailed particle microphysics (i.e. nucleation or coagulation of particles) beyond a simplified assumption of sulfuric acid ($H_2SO_{4(g)}$) nucleation for the further calculation of semi-volatile gas-phase species condensation. In this sense, the gas-phase $H_2SO_{4(g)}$ formed through oxidation of $SO_2$ can partition to the aerosol phase through both condensation and a simplified nucleation approximation. The latter consists in the assumption that 50% of the remaining $H_2SO_{4(g)}$ left in the atmosphere after solving the formation of aqueous sulfate can

nucleate and contribute to the sulfate aerosol mass. This is a simplified approach in the absence of a detailed microphysics scheme in the model. It is important to note that similar simplifications in sulfate formation processes can be found in other mass-based models in the literature. For instance, GOCART (Chin et al., 2000) and IFS-AER (Rémy et al., 2019) models accounts for sulfate formation through a single kinetic reaction that directly oxidizes $SO_2$ to form particulate sulfate.

50 Although nucleation can play an important role in determining particle number concentrations, its contribution to aerosol mass formation can be considered of secondary order (Paasonen et al., 2012). Since our study focuses on the description of aerosol mass rather than number concentration, and the resulting particulate sulfate from both processes are represented in the same size bin (i.e. fine sulfate), a detailed nucleation parameterization would not significantly impact the final results and findings of our work.

55 Previous studies, such as Kulmala et al. (1998), Vehkamäki et al. (2002) and Kuang et al. (2008), have shown that number concentration of particulate sulfate resulting from different nucleation schemes can vary in several orders of magnitude. In this context, our simplified assumption is an attempt to acknowledge the role of nucleation as a particle formation pathway while treating it in a simplified way considering the inherent uncertainty in sulfuric acid nucleation modeling and the scope of our current study.

60 The impact of adopting a simplified approach to account for particle formation through nucleation in our study may have a very limited effect. It must be taken into account that the ISORROPIA-II thermodynamic model used in the simulations to form particulate nitrate does not make a distinction between $H_2SO_{4(g)}$ gas and particulate sulfate. Thus, the sulfate phase distribution does not directly influence the thermodynamic calculation and, consequently, the formation of nitrate or ammonium in our model. Furthermore, regardless of the initial nucleation fraction assumed, most of the $H_2SO_{4(g)}$ eventually condenses

65 into particulate sulfate, either in the fine or in the coarse mode. In fact, when compared to nucleation rates from Kuang et al. (2008), our 50% assumption results in sulfate particle formation in the lower range for most $H_2SO_{4(g)}$ ambient concentrations.

Moreover, it is also important to emphasize that, due to the employment of the metastable mode in ISORROPIA-II for the thermodynamic equilibrium calculation, the results are not sensitive to the phase (solid, liquid or gas) of the sulfate input into the thermodynamic model. Consequently, if a different nucleation fraction would be applied, this would not affect the final

70 partitioning of nitrate and ammonium to the aerosol phase.

*Section 2.4:* (1) *Does the model track separately the different SO4-SS, SO4-DU, NO3-SS, NO3-DU, NH4-SS, and NH4-DU species calculated by ISORROPIA?* (2) *How many (additional) species does the model use for the different sensitivity simulations?* (3) *How much does the computational cost increase depending on the simulation setup?*

75

We acknowledge that some technical details as the ones highlighted by the reviewer were omitted in the original text. We clarify them in the following lines, and include the text added in the revised version of the manuscript to address the reviewer's questions where specified.

(1) The model does not track particle formation on dust and sea-salt separately. This is a consequence of assuming internally-
80 mixed and metastable conditions in the thermodynamic equilibrium calculation. These two assumptions lead to the tracing of total particulate $NO_3^-$, $NH_4^+$ and $SO_4^{2-}$, but prevents the individual tracing of species formed onto dust and SS particles. Nevertheless, sensitivity tests were performed to assess the relative contribution of dust and SS non-volatile cations on nitrate and ammonium formation (i.e. HYB_duUPTK, HYB_du-ssUPTK, DBCLL_noAlk, DBCLL_duAlk and DBCLL_du-ssAlk).

We revised the original text to mention this aspect in Section 2.2 of the revised manuscript (lines 195-197) as follows:
85 "The new bins account for the total mass of $NO_3^-$, $NH_4^+$, and $SO_4^{2-}$ formed on both dust and SS particles indiscriminately. Sensitivity tests, with and without dust and SS in the UPTK and TEQ processes, assess their relative contributions (see Section 2.4)."

(2) No additional species are added to the model for the sensitivity tests, just additional bins to the preexisting species $NO_3^-$,
90 $NH_4^+$ and $SO_4^{2-}$ available in the previous version of MONARCH were added, as just mentioned above.

(3) Concerning the differences in computational time between configurations, similar computational cost is found between the different heterogeneous chemistry schemes discussed with a standard deviation of 5% among them. This variation is within

the expected variability registered in the HPC employed in our study. Consequently, the analysis of the computational cost does not highlight a relevant increase in computational burden when using more complex dust heterogeneous chemistry.

We included a brief comment to this observation in the Conclusions section of the revised manuscript (line 1039) as follows:
"It is important to note that our computational cost analysis reveals highly similar processing times across the sensitivity runs, with a standard variation of only 5%. This is within the estimated variability of the supercomputing resources utilized for the present work. Consequently, the computational cost does not indicate a clear advantage in efficiency for any of the methodologies assessed."

*Section 2.4: According to Table 1, the model does not consider any heterogeneous chemistry that can promote $NO_2$ to $HNO_{3(g)}$ conversion on the surface of dust particles. How might this impact the findings of this study?*

$NO_2$ transformation to $HNO_{3(g)}$ through a surface reaction on dust is not included in MONARCH due to its relatively low relevance compared to, for example, $N_2O_{5(g)}$ hydrolysis (Underwood et al.; Jordan et al., 2003; Li et al., 2024). For instance, Jacob (2000) estimate the reaction probability after uptake of $NO_2$ onto dust to be in the order of $1 \cdot 10^{-4}$ (Liao et al., 2003), Underwood et al. estimates it to be on average $4.4 \cdot 10^{-5}$ and $2.0 \cdot 10^{-4}$, and Zhu et al. (2010) to be $2.1 \cdot 10^{-6}$. In most of the cases, the reaction rate of $NO_2$ is below the estimated average rates for $HNO_{3(g)}$ and $N_2O_{5(g)}$, ranging from $6 \cdot 10^{-4}$ to 0.1 (Fairlie et al., 2010; Zhu et al., 2010).

Including the oxidation of $NO_2$ to $HNO_{3(g)}$ on the surface of dust particles might cause a decrease in the rates of $NO_2$ conversion to $N_2O_{5(g)}$, consequently reducing the $HNO_{3(g)}$ production from the aqueous dissociation of $N_2O_{5(g)}$. Although the oxidation of $NO_2$ leads to additional $HNO_{3(g)}$, the concentrations of the latter would decrease during night-time due to the above-mentioned decrease in $N_2O_{5(g)}$ hydrolysis, mostly happening during night-time and more efficiently than the production of $HNO_{3(g)}$ from $NO_2$ oxidation (Seisel et al., 2005; Li et al., 2024; Milousis et al., 2024). Therefore, we could expect an overall decrease in $HNO_{3(g)}$ concentrations, ultimately leading to a slight reduction of particulate nitrate formation.

Nevertheless, we value the suggestion of the reviewer and we will consider adding to the model the particle surface conversion of $NO_2$ to $HNO_{3(g)}$ in future developments. We have included a brief comment on this point in Section 2.2.1 of the revised manuscript (line 216) as follows:
" No additional heterogeneous chemistry, such as the transformation of $NO_{2(g)}$ to $HNO_{3(g)}$ on the surface of dust particles, was considered due to its relatively low significance (Jacob, 2000; Jordan et al., 2003; Liao et al., 2003; Li et al., 2024).. "

*Lines 438-444 and lines 449-451: They both seem like repetitions of Sect. 2.4. The same is also happening in other parts, especially in the introduction paragraphs. I don't think this is 100% necessary, but in general, I would suggest that the authors consider ways to simplify the text of the paper. This would reduce the density of the article's information in the main body of the text.*

We concur with the reviewer's suggestion. Consequently the original text in Section 3 has been revised to enhance readability as much as possible. Most experiment definitions in the text have been removed and replaced with corresponding references, while certain numerical results, deemed less critical to the main focus of each section, have also been omitted.

*Lines 392-407: This part might be better moved to Sect. 2.1 or have another section added, as it disrupts the discussion of the different assumptions applied.*

We agree with this observation and have split the original Section 2.4 into "Section 2.4: Sensitivity runs" and "Section 2.5: Experimental setup".

*Line 667: According to the text, Karydis et al. (2016) did not apply the metastable assumption as in this work. How might this impact the difference in biases?*

Thank you for raising this point. We did not include any comment regarding the implications of the metastable assumption for our results in the original manuscript.

Based on Ansari and Pandis (2000), Karydis et al. (2016, 2021) and Milousis et al. (2024), at global scale the metastable assumption provides slightly more acidic particles and lower nitrate formation rates than the stable approach. These differences are substantially enhanced at regional scales, particularly close to arid areas that represent relevant dust sources. Regional variations also depend on the availability of nitric acid, local relative humidity and sulfate-to-nitrate ratios, which might impact fine and coarse nitrate formation differently. This is because fine nitrate principally forms from NH4+ and $HNO_{3(g)}$ neutralization, while coarse nitrate forms from crustal species neutralization, more abundant on coarse dust.

Given that the global impacts of using stable versus metastable do not seem to be significant, shifting to the stable assumption could minimally increase pH and coarse nitrate formation over arid areas, particularly over East Asia. This might lead to higher coarse nitrate formation rates and a potential worse fit of the DBCLL mechanism with observations. However, the effect on fine nitrate is difficult to predict, as it may decrease in response to higher coarse nitrate formation rates due to lower $HNO_{3(g)}$ availability. Additionally, local factors such as the availability of $NH_{3(g)}$, $HNO_{3(g)}$, $H_2SO_{4(g)}$, and relative humidity might influence the overall partitioning between fine and coarse nitrate. This could eventually alter the total nitrate evaluation.

We introduced this discussion in the revised manuscript (lines 281-286) as follows:

"At global scales, these differences noted slightly higher pH values (0.5) and nitrate formation (2%) when using the metastable assumption (Karydis et al., 2016, 2021; Milousis et al., 2024), although these differences are reported to be more important (<2 pH units and <60% nitrate concentrations) close to regions with low RH and high concentration of crustal species, or their downwind areas. However, given the global scale scope of the present study, we used the metastable assumption since it allows for full traceability of total aerosol nitrate, ammonium and sulfate formation (reactions R6-12 in Table 5)."

**Technical Comments**

*Line 461: arosol ⟶ aerosol*
*Line 868: hydrolisis ⟶ hydrolysis*

These corrections were amended in the revised manuscript.

**References**

Ansari, A. S. and Pandis, S. N.: The effect of metastable equilibrium states on the partitioning of nitrate between the gas and aerosol phases, 34, 157–168, https://doi.org/10.1016/S1352-2310(99)00242-3, 2000.

175 Chin, M., Rood, R. B., Lin, S.-J., Müller, J.-F., and Thompson, A. M.: Atmospheric sulfur cycle simulated in the global model GOCART: Model description and global properties, 105, 24 671–24 687, https://doi.org/10.1029/2000JD900384, _eprint: https://onlinelibrary.wiley.com/doi/pdf/10.1029/2000JD900384, 2000.

Fairlie, T. D., Jacob, D. J., Dibb, J. E., Alexander, B., Avery, M. A., van Donkelaar, A., and Zhang, L.: Impact of mineral dust on nitrate, sulfate, and ozone in transpacific Asian pollution plumes, Atmospheric Chemistry and Physics, 10, 3999–4012, https://doi.org/10.5194/acp-10-3999-2010, 2010.

180 Jacob, D. J.: Heterogeneous chemistry and tropospheric ozone, 2000.

Jordan, C. E., Dibb, J. E., Anderson, B. E., and Fuelberg, H. E.: Uptake of nitrate and sulfate on dust aerosols during TRACE-P, 108, https://doi.org/10.1029/2002JD003101, 2003.

Karydis, V. A., Tsimpidi, A. P., Pozzer, A., Astitha, M., and Lelieveld, J.: Effects of mineral dust on global atmospheric nitrate concentrations, 185 Atmospheric Chemistry and Physics, 16, 1491–1509, https://doi.org/10.5194/acp-16-1491-2016, 2016.

Karydis, V. A., Tsimpidi, A. P., Pozzer, A., and Lelieveld, J.: How alkaline compounds control atmospheric aerosol particle acidity, Atmos. Chem. Phys, 21, 14 983–15 001, https://doi.org/10.5194/acp-21-14983-2021, 2021.

Kuang, C., McMurry, P. H., McCormick, A. V., and Eisele, F. L.: Dependence of nucleation rates on sulfuric acid vapor concentration in diverse atmospheric locations, 113, https://doi.org/10.1029/2007JD009253, _eprint: 190 https://onlinelibrary.wiley.com/doi/pdf/10.1029/2007JD009253, 2008.

Kulmala, M., Laaksonen, A., and Pirjola, L.: Parameterizations for sulfuric acid/water nucleation rates, Journal of Geophysical Research Atmospheres, 103, 8301–8307, https://doi.org/10.1029/97JD03718, 1998.

Li, X., Yu, Z., Yue, M., Liu, Y., Huang, K., Chi, X., Nie, W., Ding, A., Dong, X., and Wang, M.: Impact of mineral dust photocatalytic heterogeneous chemistry on the formation of the sulfate and nitrate: A modelling study over East Asia, 316, 120 166, 195 https://doi.org/10.1016/j.atmosenv.2023.120166, 2024.

Liao, H., Adams, P. J., Chung, S. H., Seinfeld, J. H., Mickley, L. J., and Jacob, D. J.: Interactions between tropospheric chemistry and aerosols in a unified general circulation model, Journal of Geophysical Research: Atmospheres, 108, https://doi.org/10.1029/2001jd001260, 2003.

Milousis, A., Tsimpidi, A. P., Tost, H., Pandis, S. N., Nenes, A., Kiendler-Scharr, A., and Karydis, V. A.: Implementation of the ISORROPIA-lite aerosol thermodynamics model into the EMAC chemistry climate model (based on MESSy v2.55): implications for aerosol composi-200 tion and acidity, Geoscientific Model Development, 17, 1111–1131, https://doi.org/10.5194/gmd-17-1111-2024, 2024.

Paasonen, P., Olenius, T., Kupiainen, O., Kurtén, T., Petäjä, T., Birmili, W., Hamed, A., Hu, M., Huey, L. G., Plass-Duelmer, C., Smith, J. N., Wiedensohler, A., Loukonen, V., McGrath, M. J., Ortega, I. K., Laaksonen, A., Vehkamäki, H., Kerminen, V.-M., and Kulmala, M.: On the formation of sulphuric acid – amine clusters in varying atmospheric conditions and its influence on atmospheric new particle formation, 12, 9113–9133, https://doi.org/10.5194/acp-12-9113-2012, 2012.

205 Rémy, S., Kipling, Z., Flemming, J., Boucher, O., Nabat, P., Michou, M., Bozzo, A., Ades, M., Huijnen, V., Benedetti, A., Engelen, R., Peuch, V.-H., and Morcrette, J.-J.: Description and evaluation of the tropospheric aerosol scheme in the European Centre for Medium-Range Weather Forecasts (ECMWF) Integrated Forecasting System (IFS-AER, cycle 45R1), 12, 4627–4659, https://doi.org/10.5194/gmd-12-4627-2019, 2019.

Seisel, S., Börensen, C., Vogt, R., and Zellner, R.: Kinetics and mechanism of the uptake of N2O5 on mineral dust at 298 K, Atmospheric 210 Chemistry and Physics, 5, 3423–3432, https://doi.org/10.5194/acp-5-3423-2005, 2005.

Underwood, G. M., Song, C. H., Phadnis, M., Carmichael, G. R., and Grassian, V. H.: Heterogeneous reactions of NO2 and HNO3 on oxides and mineral dust: A combined laboratory and modeling study, 106, 18 055–18 066, https://doi.org/10.1029/2000JD900552, _eprint: https://onlinelibrary.wiley.com/doi/pdf/10.1029/2000JD900552.

Vehkamäki, H., Kulmala, M., Napari, I., Lehtinen, K. E., Timmreck, C., Noppel, M., and Laaksonen, A.: An improved parameterization for 215 sulfuric acid-water nucleation rates for tropospheric and stratospheric conditions, 107, AAC 3–10, https://doi.org/10.1029/2002JD002184, 2002.

Zhu, S., Butler, T., Sander, R., Ma, J., and Lawrence, M. G.: Impact of dust on tropospheric chemistry over polluted regions: a case study of the Beijing megacity, 10, 3855–3873, https://doi.org/10.5194/acp-10-3855-2010, 2010.

**REFEREE #3**

**Summary**

*Nitrogen is important in several perspectives, and it can affect air quality mainly through oxidized and reduced nitrogen. This study focuses on the nitrate formation processes, and designed a series of numerical experiments to elucidate the main governing mechanisms. The overall study is important and interesting, but I do feel the manuscript needs a substantial revision to be easily followed.*

We appreciate the reviewer's comprehensive revision and insightful comments, which we think point to important aspects of the paper that deserve attention. We address the comments in the lines below. Additionally, the original text of the manuscript has been substantially revised to ease its readability and comprehension, as outlined in the responses that follow.

**Comments**

*1. Abstract: regarding the first sentence. I am not sure the main focus is nitrate or the desert dust. I feel the authors try to elucidate the issues of nitrate, and dust is one of the factors affecting concentrations of nitrate. Line 5-6: "This study investigates key processes in nitrate formation over dust and evaluates their representation in models." By reading the manuscript, I think the authors not only examine nitrate formation over dust, but also on sea salt and others. Does the abstract correctly deliver the message?*

We understand this comment and it was indeed a topic of discussion among the authors of the manuscript.
While our study incorporates nitrate formation over sea salt and other aerosol species, these are included primarily because they represent crucial pathways to accurately reproduce global particulate nitrate observations. However, the main scope of the work is to investigate the role of dust and its representation in models for nitrate formation. Through systematic sensitivity simulations, we aim to better understand the pathways driving nitrate formation on dust. Some sensitivity simulations - such as the inclusion or exclusion of nitrate formation on sea salt in some mechanisms - are intended to evaluate the role of sea salt in the global nitrate burden, given its competition with dust for gas-phase species. However, these sensitivity simulations do not aim at specifically investigating the mechanisms governing nitrate formation on sea salt.
For these reasons, we emphasize the role of dust representation in the formation of nitrate in several parts of the manuscript, including the abstract, instead of referring to a more general nitrate formation analysis and discussion. The scope of the work is better clarified in the revised manuscript.

*2. Introduction: the authors have tried to discuss the potential issues related to heterogeneous reactions (e.g., $HNO_{3(g)}$ to dust). Similar as the previous question: I am not sure the main focus of this study is dust or nitrate. I feel it is nitrate instead of desert dust. At the third last paragraph, the authors pointed out a general inaccuracy of current models in reproducing nitrate and misrepresentation of nitrogen heterogeneous chemistry processes on dust and sea salt. Following this, the authors mentioned Following that, the authors try to systematically investigate the underlying processes governing the issues. However, I feel it is not clear what issues are in the current models. Is it due to the DMT or TEQ? It does not look like the authors have discussed the problems clearly. In this way, the readers would not know how the authors can disentangle the problems.*

We understand and appreciate the reviewer's concern regarding the potential ambiguity in the scope of the present study.
As clarified in the previous comment, our study aims to study the mechanisms governing nitrate formation on mineral dust, assessing its sensitivity to critical variations in nitrate formation pathways on these species. While some sensitivity simulations intend to assess the relative role of, for instance, sea salt, they do not explore the mechanisms governing the formation on other

45    species.

Regarding the clarity in the presentation of the issues faced by current models, we appreciate the reviewer's perspective on this matter.

For this study we conducted a comprehensive literature review to understand and contextualize existing dust chemistry
50   mechanisms incorporated in atmospheric models, as it is included in the introduction section. This review served to understand that very often, difficulties faced by models when reproducing global particulate nitrate observations could not be attributed solely to their heterogeneous dust chemical mechanism (i.e. TEQ, HYB, DBCLL or DMT), but also to a wide variety of other factors, such as the emission inventories, the alkalinity representation of dust and sea-salt or their size distribution parameterization, among many others. Indeed, the AeroCom Phase III nitrate experiment (Bian et al., 2017) highlights the significant
55   disparities among participating models, driven by differences in the factors mentioned.

In this study, we conducted a systematic investigation of the key chemical factors influencing particulate nitrate formation on dust, utilizing a single atmospheric chemistry model with consistent transport, emissions, and chemical schemes across all experiments. Therefore, we do not delve into the specific issues faced by current models when trying to reproduce nitrate formation on dust, leaving the possibility to, in future studies, investigate other factors such as variations in dust mineralogy.

60

We have clarified the scope of our work in the Introduction (line 106 of the revised manuscript):

"The scope of the present work is to understand the role of dust in $NO_3^-$ formation through a systematic investigation of the underlying processes governing dust heterogeneous chemistry. [...] While our primary emphasis is on the heterogeneous chemistry on dust surfaces, we also account for nitrate formation on SS and its alkalinity."

65

*3. Line 154: "assuming complete nucleation" I am not sure what complete nucleation means. Does this mean that all gas $H_2SO_{4(g)}$ is nucleated? This does not seem reasonable.*

70   By "complete nucleation", we mean that all gas $H_2SO_{4(g)}$ remaining in the atmosphere after aqueous sulfate formation is assumed to partition to the aerosol phase through nucleation, as pointed out by the reviewer.

The assumption of complete nucleation for $H_2SO_{4(g)}$ is a simplification to represent $H_2SO_{4(g)}$ transfer to particulate phase in a mass-based model such as MONARCH. As the model does not include particle microphysics nor operates as a number-concentration based model, this approach ensures a first order treatment of nucleation and condensation processes of $H_2SO_{4(g)}$
75   in the model. The products of both processes are treated equally in the model, both represented within the model's fine particulate sulfate bin (Spada, 2015). Given the extremely low vapor pressure of $H_2SO_{4(g)}$ (Yu and Luo, 2009; Wang et al., 2015), it is reasonable to assume the complete partitioning of $H_2SO_{4(g)}$ into the particulate phase through nucleation and condensation. Similar approximations have also been assumed in other models, such as GoCart (Chin et al., 2000) and IFS-AER (Rémy et al., 2019), where even a more simplistic approximation is considered to account for sulfate mass production in the
80   atmosphere through a single kinetic reaction oxidizing $SO_{2(g)}$ and directly forming particulate sulfate.

To clarify the concept of complete nucleation, we introduced the following text in Section 2.1 of the revised manuscript (line 159):

"At the end of each chemistry integration time step, the remaining $H_2SO_{4(g)}$ that has not formed aqueous sulfate is assumed to fully nucleate into fine particulate $SO_4^{2-}$ (Spada, 2015)."

85

Although the assumption of complete nucleation of $H_2SO_{4(g)}$ may not be considered completely accurate, in our particular context of modeling dust heterogeneous chemistry, the phase of $H_2SO_{4(g)}$ (either solid, liquid or gas) does not influence the particulate nitrate and ammonium formation from the thermodynamic equilibrium calculation. This is because the ISORROPIA-II thermodynamic model uses total sulfate - indistinctively of its phase - as input to compute the nitrate partitioning. Furthermore,
90   since in our work we assume metastable conditions with ISORROPIA-II, it does not consider the precipitation of solid particles from $H_2SO_{4(g)}$ condensation. Consequently, the thermodynamic equilibrium calculation does not affect the partitioning of $H_2SO_{4(g)}$, and we assume its complete condensation into the respective size mode. This consideration would change if we

employed ISORROPIA-II in the stable mode, where solids from the partitioning of $H_2SO_{4(g)}$ (such as $CaSO_4$) would need to be explicitly accounted as outputs.

95

*4. Line 181: "In this study, we investigate the primary chemical pathways responsible for NO3- formation on coarse particles by integrating" Why do the authors only investigate the pathways of nitrate formation on coarse particles? How about the fine mode?*

100

We concur with the reviewer in the possible confusion that this sentence might lead to. We modified the sentence to try to improve its clarity to (Section 2.2 line 188 of the revised manuscript):
"In this study, we investigate the primary chemical pathways responsible for $NO_3^-$ formation on preexisting particles, with a particular focus on coarse dust particles, by integrating mechanisms of varying complexity within the global model MONARCH"

105

We consider it is important to note that particular focus is put in the role of "coarse particles" since most of the sensitivity tests are performed varying the mechanisms for coarse nitrate formation particularly on corse dust particles. Conversely, fine nitrate formation is consistently formed through TEQ in all mechanisms analyzed here.

110

*5. Line 188 hydrophilic Why hydrophilic? Some explanation is needed.*

We agree in that the text should provide a better explanation of this point. The additional bin is considered to be hydrophilic because the studied formation mechanisms occur under moist conditions, a regime in which the nitrate formation pathways
115   are more efficient. For instance, the uptake coefficient on dust as a function of relative humidity is parameterized for non-zero humidity values. Additionally, thermodynamic equilibrium is performed under the metastable solution assumption (i.e. considering only gas and liquid species). Therefore it requires a positive relative humidity to account for the deliquescence of crustal species.

120   We added a clarification as follows (Section 2.2 line 193 in the revised manuscript):
"To trace the formation of fine and coarse $NO_3^-$, $NH_4^+$, and $SO_4^{2-}$ under moist conditions — the primary regime where these formation pathways occur (Usher et al., 2003; Jordan et al., 2003) — an additional hydrophilic bin for the coarse mode of these species is added to the default MONARCH size parametrization, as detailed in Supplementary Table S2."

125

*6. Table 1: Not sure why in this table, there is no uptake of $HNO_{3(g)}$ on dust. Indeed, Line 214: the authors mentioned that "For the uptake of $HNO_{3(g)}$ on dust (R4-5 in Table 1)". I don't understand why R5 is related to dust (it apparently says sea salt). The caption of Table 1 says DU represents Dust, why specifically using $CaCO_3$?*

130   We thank the reviewer for highlighting this point. The sentence in line 214 has a typo and should only refer to reaction R4. We have amended the sentence in the revised manuscript.
In Table 1, the uptake of $HNO_{3(g)}$ on dust is expressed in R4. The explanation why we used $CaCO_3$ instead of Dust is addressed in the note 3 of Table 1. For this reaction, we scale the dust uptake coefficient to the actual $CaCO_3$ content to account for alkalinity. Consequently, the uptake is considered to happen only on $CaCO_3$ particles. This is not the case for reactions R1
135   and R3. In the case of $SO_{2(g)}$ (R3), the uptake coefficient is also scaled for $CaCO_3$ content with *Sc*. However, an exception is made here, setting *Sc* to 1.0 in the case without alkalinity. That is why in Table 1 reaction R3, DU is specified instead of $CaCO_3$. This was mentioned in Note 2 of Table 1, but we clarify the explanation of the note as follows:
"That is why this reaction is considered to happen over DU(aer) and not only over CaCO3(aq), although it assumes same dust alkalinity as in nitric acid uptake (R4).

140

*7. Line 216-218: "However, literature reports varying values for Sc based on dust alkalinity assumptions, ranging from Sc= 1/30 for the industrially-standardized Arizona Test Dust (Möhler et al., 2006; Herich et al., 2009; Suman et al., 2024) to Sc=0.018 for samples from the China Loess (with 39% $CaCO_3$ content) (Krueger et al., 2004; Wei, 2010)." What is the setting in this study considering the large range of Sc?*

The values adopted for *Sc* in our study are specified in Line 224 of the original manuscript: "Sc = 1.80 and Sc = 1.52 for the two alkalinity values used in our experiments". It is important to note that we use the alkalinity scaling factor *Sc* to normalize the Fairlie et al. (2010) values, which assumes a different non-volatile cation content than our work. This is explained in Lines 219-225 in the original manuscript. We did not evaluate alternative *Sc* values, given that our approach considers a constant dust mineralogical composition.

*8. For RH above 90%, $\gamma(SO_{2(g)})$ remains constant at $\gamma(SO_{2(g)})$ = Sc·5.0·10-4 What is the value of Sc in this equation*

In this case, *Sc* has the same value as the $HNO_{3(g)}$ case, $Sc = 1.80$ for Journet et al. (2014) and $Sc = 1.52$ for Claquin et al. (1999) mineral datasets, as mentioned in Lines 224 and 229 and Section 2.2.1 in the original manuscript.

However, the uptake reaction of $SO_{2(g)}$ is performed even in the absence of alkalinity, where *Sc* is set to 1.0 and not 0.0, as it was assumed for $HNO_{3(g)}$ uptake in the absence of alkalinity (see note 2 in Table 1).

We clarified the original sentence in the revised manuscript as follows (Section 2.2.1 line 241):

"The same *Sc* values as used for $\gamma(HNO_3)$ are applied for $\gamma(SO_2)$: $Sc = 1.80$ and $Sc = 1.52$. However, if alkalinity is not considered, *Sc* is set to 1.0 (and not zero, as the case for $\gamma(HNO_3)$) to account for the oxidation of $SO_{2(g)}$ by deliquesced $O_3$ and $NO_2$ (Usher et al., 2002; Prince et al., 2007; Yu et al., 2017; Li et al., 2024). For RH above 90%, $\gamma(SO_2)$ remains constant at $\gamma(SO_2) = Sc \cdot 5.0 \cdot 10^{-4}$."

*9. Line 301: The dust NVC global average content result in: 5.17% Ca2+, 0.79% Na+, 2.37% K+, 1.32% Mg2+ for the Journet et al. (2014) dataset, and 3.68% Ca2+, 0.87% Na+, 3.15% K+, 1.75% Mg2+ for Claquin et al. (1999). Which one does the author use? Please add some descriptions besides of laying out the two different values.*

Table 3 column DU Alkalinity specifies which dataset is used in the different sensitivity runs discussed in our work. Global average values derived from Journet et al. (2014) are employed in all runs except the case DBCLL_ClaqAlk, where Claquin et al. (1999) dataset is used. With the exception of the DBCLL_ClaqAlk run, all the others experiments use 5.17% Ca2+, 0.79% Na+, 2.37% K+, 1.32% Mg2+ NVC percentages, derived from Journet et al. (2014).

This is now clarified in the revised manuscript Section 2.2.3 lines 319-323 adding a brief description of the main differences between the two datasets as follows:

"In most of the sensitivity runs, the Journet et al. (2014) global average is employed, if not stated otherwise (see Table ??). Values for Claquin et al. (1999) are used solely in one sensitivity test, as explained in Section ??. These values are within the range reported by Karydis et al. (2016) ($5.36\pm3.69\%$ $Ca^{2+}$, $2.46\pm1.90\%$ $Na^+$, $2.08\pm1.34\%$ $K^+$, $1.96\pm2.20\%$ $Mg^{2+}$). The Journet et al. (2014) dataset results in a higher proportion of $Ca^{2+}$ compared with Claquin et al. (1999), while similar fractions for $Na^+$ and $K^+$ are reported."

*10. Table 4: there is no unit*

We thank the reviewer for pointing out the missing units in Table 4. We have amended this by specifying the units in the Table's caption. Both bias and rmse are reported in $\mu g \ m^{-3}$.

190    *11. Descriptions in section 3: I feel it is not easy to follow. Too many numbers. Too many abbreviations make it very hard to follow. If possible, please elaborate a bit more to emphasize the main focus. More references can be added to enhance the readability, and readers might know what is important.*

We value this observation and agree with this comment. The text of this section has been significantly revised and reduced
195    to ease the reading and highlighting the main scope and findings of our results. We simplified the definitions, the use of abbreviations, and the numerical results provided in the original manuscript that were not relevant for the main findings of each subsection. We also moved some experiments' analyses to the Supplementary material (Section S5). We believe that the revised section addresses now the concerns raised by the reviewer.

200

*12. The authors mentioned many times about dust and sea salt. Did the authors do any evaluation on these two species? Can the model reasonably reproduce dust and sea salt?*

The present work does not specifically evaluate dust nor sea-salt concentrations. For dust, this study adopts the same scheme
205    evaluated in Klose et al. (2021) and Gonçalves Ageitos et al. (2023), in both studies a comprehensive evaluation of the dust cycle were provided. Concerning the sea-salt, our simulations used the scheme already evaluated in Spada et al. (2013) where different parameterizations were compared and extensively evaluated. In our work, we report dust and sea-salt column loads in Supplement Figures S10 and S11 to support the discussion on nitrate formation and focus the main evaluation to nitrate and particulate matter.
210    To address the reviewer comment, we have added the following sentence on the assessment of the dust and sea salt life cycle simulated by the model citing the mentioned references (Section 2.1 lines 185-186 of the revised manuscript):
"For a detailed evaluation of the model's dust cycle, the reader is referred to Klose et al. (2021) and Gonçalves Ageitos et al. (2023), and for SS to Spada et al. (2013)."

215

*13. For the reduced and oxidized nitrogen, did the authors compare the results with observations? I think comparison with observations are useful to understand the nitrogen budget.*

We value this recommendation and agree that this evaluation would be useful. Consequently, we added the evaluation of
220    reduced and oxidized species in the Supplementary Section S6.
We discussion of the evaluation is also included now in Section 3.3.2 (lines 844-848 of the revised manuscript) as follows:
"The observational evaluation of both reduced and oxidized species, along with the monitoring stations used, can be found in the Supplementary Section S6. Overall, a general good agreement with observations is obtained. Oxidized nitrogen species exhibit low biases in all mechanisms except HYB_du-ssUPTK, which overestimates observations throughout the studied pe-
225    riod, supporting the conclusion that UPTK coefficients used are excessively efficient. Reduced nitrogen species, while slightly underestimated, remain well within the observational variability range."

*14. The conclusions: It seems not clear what is the best option to take for the nitrate reactions. The reversible or irreversible?*
230    *Please summarize to make it clear whether there is an optimal option, or multiple options to derive reasonable simulations of nitrate.*

We thank the reviewer for pointing this out and agree that our manuscript did not provide a clear recommendation regarding the most suitable scheme for simulating nitrate formation. Based on our analysis and evaluations, we recommend adopting
235    the reversible representation in models, as it enables a more detailed description of key processes, such as alkalinity in nitrate partitioning, without significantly increasing the computational cost.
We have introduced a statement in the Conclusions (line 1030) to highlight the best mechanism based on our analysis as follows:

"Overall, the DBCLL_du-ssAlk scheme demonstrates the best accuracy compared to the other tested configurations, as evidenced by its closer alignment with observations."

**References**

Bian, H., Chin, M., Hauglustaine, D. A., Schulz, M., Myhre, G., Bauer, S. E., Lund, M. T., Karydis, V. A., Kucsera, T. L., Pan, X., Pozzer, A., Skeie, R. B., Steenrod, S. D., Sudo, K., Tsigaridis, K., Tsimpidi, A. P., and Tsyro, S. G.: Investigation of global particulate nitrate from the AeroCom phase III experiment, Atmospheric Chemistry and Physics, 17, 12 911–12 940, https://doi.org/10.5194/acp-17-12911-2017, 2017.

Chin, M., Rood, R. B., Lin, S.-J., Müller, J.-F., and Thompson, A. M.: Atmospheric sulfur cycle simulated in the global model GOCART: Model description and global properties, 105, 24 671–24 687, https://doi.org/10.1029/2000JD900384, _eprint: https://onlinelibrary.wiley.com/doi/pdf/10.1029/2000JD900384, 2000.

Claquin, T., Schulz, M., and Balkanski, Y. J.: Modeling the mineralogy of atmospheric dust sources, Journal of Geophysical Research: Atmospheres, 104, 22 243–22 256, https://doi.org/10.1029/1999JD900416, 1999.

Gonçalves Ageitos, M., Obiso, V., Miller, R. L., Jorba, O., Klose, M., Dawson, M., Balkanski, Y., Perlwitz, J., Basart, S., Di Tomaso, E., Escribano, J., Macchia, F., Montané, G., Mahowald, N. M., Green, R. O., Thompson, D. R., and Pérez García-Pando, C.: Modeling dust mineralogical composition: sensitivity to soil mineralogy atlases and their expected climate impacts, Atmospheric Chemistry and Physics, 23, 8623–8657, https://doi.org/10.5194/acp-23-8623-2023, 2023.

Jordan, C. E., Dibb, J. E., Anderson, B. E., and Fuelberg, H. E.: Uptake of nitrate and sulfate on dust aerosols during TRACE-P, 108, https://doi.org/10.1029/2002JD003101, 2003.

Journet, E., Balkanski, Y., and Harrison, S. P.: A new data set of soil mineralogy for dust-cycle modeling, Atmospheric Chemistry and Physics, 14, 3801–3816, https://doi.org/10.5194/ACP-14-3801-2014, 2014.

Karydis, V. A., Tsimpidi, A. P., Pozzer, A., Astitha, M., and Lelieveld, J.: Effects of mineral dust on global atmospheric nitrate concentrations, Atmospheric Chemistry and Physics, 16, 1491–1509, https://doi.org/10.5194/acp-16-1491-2016, 2016.

Klose, M., Jorba, O., Ageitos, M. G., Escribano, J., Dawson, M. L., Obiso, V., Tomaso, E. D., Basart, S., Pinto, G. M., Macchia, F., Ginoux, P., Guerschman, J., Prigent, C., Huang, Y., Kok, J. F., Miller, R. L., and García-Pando, C. P.: Mineral dust cycle in the Multiscale Online Non-hydrostatic AtmospheRe CHemistry model (MONARCH) Version 2.0, Geosci. Model Dev, 14, 6403–6444, https://doi.org/10.5194/gmd-14-6403-2021, 2021.

Li, X., Yu, Z., Yue, M., Liu, Y., Huang, K., Chi, X., Nie, W., Ding, A., Dong, X., and Wang, M.: Impact of mineral dust photocatalytic heterogeneous chemistry on the formation of the sulfate and nitrate: A modelling study over East Asia, 316, 120 166, https://doi.org/10.1016/j.atmosenv.2023.120166, 2024.

Prince, A. P., Kleiber, P., Grassian, V. H., and Young, M. A.: Heterogeneous interactions of calcite aerosol with sulfur dioxide and sulfur dioxide–nitric acid mixtures, Physical Chemistry Chemical Physics, 9, 3432–3439, https://doi.org/10.1039/B703296J, 2007.

Rémy, S., Kipling, Z., Flemming, J., Boucher, O., Nabat, P., Michou, M., Bozzo, A., Ades, M., Huijnen, V., Benedetti, A., Engelen, R., Peuch, V.-H., and Morcrette, J.-J.: Description and evaluation of the tropospheric aerosol scheme in the European Centre for Medium-Range Weather Forecasts (ECMWF) Integrated Forecasting System (IFS-AER, cycle 45R1), Geoscientific Model Development, 12, 4627–4659, https://doi.org/10.5194/gmd-12-4627-2019, 2019.

Spada, M.: Development and Evaluation of an Atmospheric Aerosol Module Implemented Within the Nmmb/Bsc-Ctm Michele Spada Phd Thesis 2015, 2015.

Spada, M., Jorba, O., Pérez García-Pando, C., Janjic, Z., and Baldasano, J. M.: Modeling and evaluation of the global sea-salt aerosol distribution: sensitivity to size-resolved and sea-surface temperature dependent emission schemes, 13, 11 735–11 755, https://doi.org/10.5194/acp-13-11735-2013, 2013.

Usher, C. R., Al-Hosney, H., Carlos-Cuellar, S., and Grassian, V. H.: A laboratory study of the heterogeneous uptake and oxidation of sulfur dioxide on mineral dust particles, Journal of Geophysical Research: Atmospheres, 107, https://doi.org/10.1029/2002JD002051, 2002.

Usher, C. R., Michel, A. E., and Grassian, V. H.: Reactions on Mineral Dust, Chemical Reviews, 103, 4883–4939, https://doi.org/10.1021/cr020657y, 2003.

Wang, Z. B., Hu, M., Pei, X. Y., Zhang, R. Y., Paasonen, P., Zheng, J., Yue, D. L., Wu, Z. J., Boy, M., and Wiedensohler, A.: Connection of organics to atmospheric new particle formation and growth at an urban site of Beijing, Atmospheric Environment, 103, 7–17, https://doi.org/10.1016/j.atmosenv.2014.11.069, 2015.

Yu, F. and Luo, G.: Simulation of particle size distribution with a global aerosol model: contribution of nucleation to aerosol and CCN number concentrations, Atmos. Chem. Phys, 9, 7691–7710, www.atmos-chem-phys.net/9/7691/2009/, 2009.

Yu, Z., Jang, M., and Park, J.: Modeling atmospheric mineral aerosol chemistry to predict heterogeneous photooxidation of SO$_2$, 17, 10 001–10 017, https://doi.org/10.5194/acp-17-10001-2017, 2017.